# Have Large Language Models Learned to Reason?
# A Characterization via 3-SAT

**Rishi Hazra[1], Gabriele Venturato[2], Pedro Zuidberg Dos Martires[1] & Luc De Raedt[1,2]**
[1]Centre for Applied Autonomous Sensor Systems (AASS),
Örebro University, 70182, Sweden
[2]Department of Computer Science, KU Leuven, 3001, Belgium
{rishi.hazra, pedro.zuidberg-dos-martires}@oru.se
{gabriele.venturato, luc.deraedt}@kuleuven.be

## Abstract

Large Language Models (LLMs) have been touted as AI models possessing advanced reasoning abilities. In theory, autoregressive LLMs with Chain-of-Thought (CoT) can perform more serial computations to solve complex reasoning tasks. However, recent studies suggest that, despite this capacity, LLMs do not truly learn to reason but instead fit on statistical features. To study the reasoning capabilities in a principled fashion, we adopt a computational theory perspective and propose an experimental protocol centered on 3-SAT – the prototypical NP-complete problem lying at the core of logical reasoning and constraint satisfaction tasks. Specifically, we examine the phase transitions in random 3-SAT and characterize the reasoning abilities of state-of-the-art LLMs by varying the inherent hardness of the problem instances. By comparing DeepSeek R1 with other LLMs, our findings reveal two key insights (1) LLM accuracy drops significantly on harder instances, suggesting all current models struggle when statistical shortcuts are unavailable (2) Unlike other LLMs, R1 shows signs of having learned the underlying reasoning. Following a principled experimental protocol, our study moves beyond the benchmark-driven evidence often found in LLM reasoning research. Our findings highlight important gaps and suggest clear directions for future research. Link to our code.

## 1 Introduction

The success and versatility of Large Language Models (LLMs) have sparked widespread interest and debate on whether LLMs are capable of reasoning. Recent studies suggest that LLMs are inherently capable of zero-shot reasoning (Kojima et al., 2022). This ability has been shown to *emerge* and improve with scale (Wei et al., 2022a; Srivastava et al., 2023), and can be further enhanced through prompting techniques that encourage LLMs to *think step-by-step* (Wei et al., 2022b; Yao et al., 2023b). Demonstrations include, inter alia, planning (Huang et al., 2022; Hazra et al., 2024b), theorem proving (Jiang et al., 2023; Welleck et al., 2022), search and optimization (Romera-Paredes et al., 2024; Hazra et al., 2024a), self-reflection (Yao et al., 2023a; Madaan et al., 2023), and tool usage (Schick et al., 2023).

Conversely, a growing body of research presents a more critical view of these reasoning abilities. For instance, LLMs may exhibit limitations in consistent logical reasoning (Arkoudas, 2023; Saparov & He, 2023), effective planning (Valmeekam et al., 2022), and accurate self-evaluation of their outputs (Stechly et al., 2023). So, to what extent can LLMs reason?

Recent works address this question by characterizing LLMs' theoretical capabilities and limitations using worst-case complexity analysis. Peng et al. (2024) demonstrated that multi-layer transformers cannot solve problems such as Derivability, 2-SAT, and Horn SAT. However, with $T$-chain of thought (CoT) steps, transformers' abilities can be extended up to those solvable by Boolean circuits of size $T$ (Li et al., 2024). Moreover, with polynomially many *correct* intermediate CoT steps, an LLM can compute all circuits in polynomial size,

P/poly, a superclass of P. Therefore, LLMs should be able to solve reasoning problems falling into these complexity classes, like 2-SAT and Horn-SAT. It is also expected to solve some – but not all – 3-SAT instances since certain problems may require circuit complexities that exceed polynomial bounds. Empirically, this aligns with test-time compute scaling studies that show improved reasoning performance with increased computational resources (Lightman et al., 2024; Snell et al., 2025). **Thus, from a theoretical standpoint, LLMs equipped with CoT possess the *capacity* to solve reasoning problems falling in P or P/Poly**.

However, despite this, recent works have shown that LLMs are **not *learning* to reason**, by fitting on statistical features and not internalizing the logic (i.e. learning to reproduce the style and not logic) (Zhang et al., 2023) much like the *Clever Hans Cheat* (Bachmann & Nagarajan, 2024). With the advent of Large Reasoning Models (LRMs) like DeepSeek R1 (DeepSeek-AI, 2025a) which leverage more test-time compute, there is renewed optimism about enhancing reasoning capabilities. However, it is unclear if this marks a real step change, due to limitations in current reasoning benchmarks like (1) growing concerns about dataset contamination[1] (Zhang et al., 2024) that can inflate the reasoning performance; (2) conflation of commonsense reasoning rooted in *retrieving* world knowledge, and logical or deductive reasoning – requiring algebraic *manipulation* of knowledge (Genesereth & Nilsson, 1987) – making it challenging to decouple the two.

To move beyond these limitations, a more principled approach to evaluating reasoning is needed. We adopt Leon Bottou's definition, which defines reasoning as "*algebraically manipulating previously acquired knowledge to answer a new question*" (Bottou, 2014). This closely aligns with Russell and Norvig's description of AI as rational thinking (Russell & Norvig, 2010). Building on this, we ask: Have LLMs learned to reason, and if so, to what extent? We answer through the following contributions:

**(1) Characterizing LLM-reasoning from a computational theory perspective.** Adhering to Leon Bottou's definition, we propose an experimental framework centered on 3-SAT, to evaluate reasoning. Introduced in Figure 1, 3-SAT is a foundational problem in computational complexity, and many problems in AI such as (propositional fragments of) logical reasoning, planning, and constraint satisfaction can be reduced to 3-SAT. We also extend our analysis to tractable fragments like 2-SAT (NL-Complete) (Bollobás et al., 2001) and 1-3 Horn-SAT (P-Complete) (Demopoulos & Vardi, 2006) which also exhibit phase transitions, revealing interesting observations across complexity classes. Our approach provides a formal and robust evaluation of reasoning abilities and bridges computational theory with modern AI.

**(2) Studying Phase Transition Characteristics of LLMs.** We investigate the phase transition characteristics of LLMs (Cheeseman et al., 1991), i.e., how LLMs' performance varies in the easy and hard regions of the problem space, unlike standard reasoning benchmarks that overlook variations due to *inherent hardness* of problems. We observe that LLM performance declines in the hard region where statistical features are largely absent.

**(3) Comprehensive evaluation of state-of-the-art LLMs.** We conducted extensive experiments across state-of-the-art open-source and proprietary LLMs. We find a significant gap between the performances of DeepSeek R1 (DeepSeek-AI, 2025a) to other LLMs like GPT-4o (OpenAI, 2024), Claude 3.7 Sonnet (Anthropic, 2025), Gemini 2.0 Flash (Google, 2024), and DeepSeek V3 (DeepSeek-AI, 2025b). Interestingly, the CoT traces produced by DeepSeek R1 reveal patterns suggestive of an in-context tree search with backtracking, which may account for its superior performance. In contrast, the other models consistently fail to demonstrate comparable reasoning capabilities. This leads us to believe that LRMs like R1 could be a **step-change in reasoning abilities**.

## 2 Preliminaries

### 2.1 Phase Transitions in Random 3-SAT

We study the reasoning capabilities of LLMs on random 3-SAT problems. 3-SAT constitutes one of the most fundamental problems in computer science as it is the prototypical

---

[1]Data closely resembling the benchmark leaks into the training data.

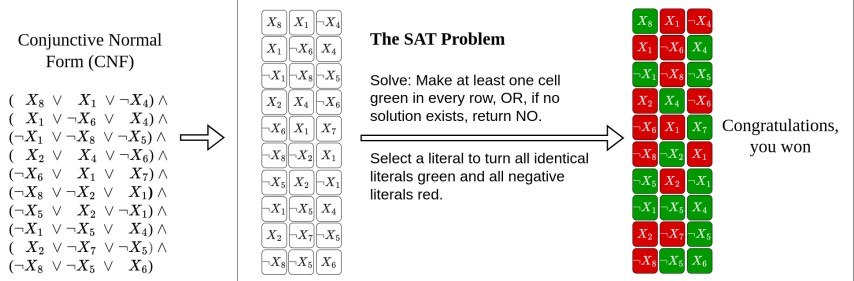

Figure 1: **The 3-SAT problem**, visualized using a variant of the SAT game (Roussel). In SAT, the goal is to return a truth assignment to Boolean variables that satisfies a Boolean formula in conjunctive normal form (CNF), or return unSAT if none exists. In the visualization, each row represents a clause—a disjunction (logical OR, $\vee$) of literals, where each literal is either positive ($X_1$) or negative ($\neg X_1$). A clause is satisfied if at least one of its literals is assigned true. Clauses are joined by logical AND ($\wedge$), so all must be satisfied for the formula to hold. If no such assignment exists, the formula is unsatisfiable.

NP-complete problem, lying at the foundation of computational complexity theory. Moreover, various prevalent reasoning problems in artificial intelligence, such as planning and constraint satisfaction, can be reduced to solving 3-SAT problems (Garey & Johnson, 1990).

By randomly sampling 3-SAT formulas[2], we avoid domain-specific biases, and it lets us control the complexity of the generated formulas. In fact, an interesting empirical observation is the presence of a *phase transition* in random 3-SAT problems (Cheeseman et al., 1991). When randomly sampling 3-SAT formulas, one can observe a sharp change in the probability of a 3-SAT formula being satisfiable when plotted against $\alpha = m/n$, where $m$ is the number of clauses and $n$ is the number of variables. For random 3-SAT, this phase transition occurs at $\alpha_c \approx 4.267$ (Mertens et al., 2006; Ding et al., 2015), i.e. the point at which a randomly sampled 3-SAT formula has equal probability to be satisfiable or unsatisfiable. This naturally divides 3-SAT problems into three regions: the under-constrained region below the threshold (Easy), the constrained region in the neighborhood of the threshold (Hard), and the over-constrained region above the threshold (Easy), cf. Figure 2.

## 2.2 SAT Solvers

SAT solvers are tools to automatically verify the satisfiability of propositional logic formulas. The Davis–Putnam–Logemann–Loveland (DPLL) algorithm (Davis et al., 1962) is a key component of modern SAT solvers. It consists of a backtracking-based algorithm, equipped with heuristics, to efficiently explore the search space and determine if a formula is satisfiable (Figure 2). The algorithm selects a literal and assigns a truth value to it. This is applied recursively until all clauses are satisfied, meaning that the original formula is satisfiable. Modern SAT solvers are based on Conflict-Driven Clause Learning (CDCL) (Silva & Sakallah, 1996) which enhances DPLL with conflict analysis and clause learning. In Figure 2, we show how the time to determine the satisfiability of a random 3-SAT formula varies in function of $\alpha$. Most prominently, we see a pronounced peak in runtime around the $\alpha_c$.

Analogously to characterizing SAT solvers by their behavior with varying $\alpha$, we study the reasoning capabilities of LLMs with respect to the phase transition in random 3-SAT.

## 3 Related Work

Merrill & Sabharwal (2023) established that multi-layer transformers belong to the complexity class log-uniform $TC^0$. Subsequently, Peng et al. (2024) demonstrated that multi-layer transformers cannot solve problems such as Derivability, 2-SAT, Horn SAT, and Circuit

---

[2]We use Selman et al. (1996) random model, where clauses are sampled with replacement.

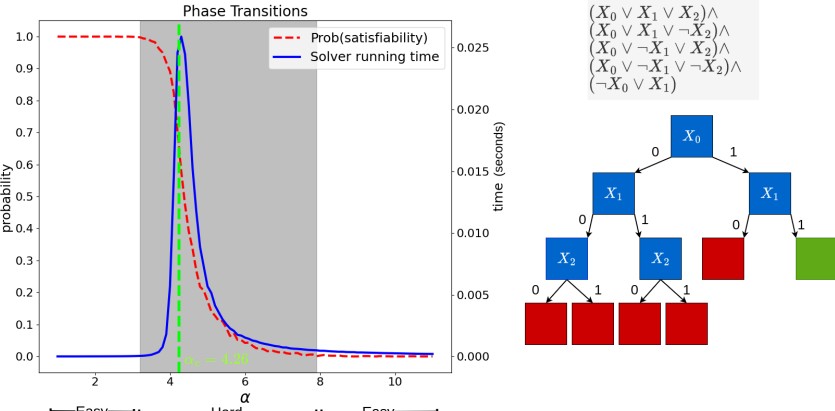

Figure 2: [Left]: **Random 3-SAT Phase Transitions** (Cheeseman et al., 1991). Plotted in red is the probability of a randomly sampled 3-SAT formula being satisfied against the hardness $\alpha$ of the formula. We can observe a clear phase transition occurring at $\alpha_c \approx 4.267$ (marked by a green - -). We identify two Easy regions, one on either side of $\alpha_c$. The gray area in the middle denotes the Hard region. The boundaries of the hard region are defined where the probability of the formula being satisfied ceases to be deterministically 1 (left) or 0 (right). The solid blue line shows the mean time taken by the MiniSAT solver to solve 3-SAT instances. Notably, there is a spike in the solver's runtime near $\alpha_c$. This is due to the absence of useful heuristics in this region, forcing the solver to resort to exhaustive searches. [Right]: **DPLL Search Trace** from a SAT Solver for the given formula. Red boxes denote unsatisfiable assignments; the green box highlights a satisfying one. 0, 1 are truth assignments.

Evaluation unless L=NL. While these results provide worst-case performance bounds for transformer architectures, their relevance to average-case complexity is limited (Coarfa et al., 2000), therefore, offering only partial insights into the practical reasoning capabilities of LLMs. Notably, recent works have shown that $T$ chain of thought (CoT) steps can extend transformers' abilities beyond $\text{TC}^0$, up to those solvable by Boolean circuits of size $T$ (Li et al., 2024). With $T$ being polynomial in the input size, LLMs can in theory solve problems in P class like 2-SAT and Horn SAT. Our SAT experiments support this, showing improved reasoning with extended CoT steps. Yet, a central question remains: *can LLMs learn to reason effectively, given that they are theoretically capable?*

To this end, Dziri et al. (2023) investigate the performance of LLMs on compositional tasks with varying levels of complexity. Their experiments reveal a significant performance decline as task complexity increases, measured by problem size and reasoning depth. The findings indicate that while models can memorize single-step operations from training, they fail to compose these steps into correct reasoning paths. Similarly, Zhang et al. (2023) found that BERT-based models fail to generalize to out-of-distribution data even within a tractable (i.e., not NP-complete) problem class, overfitting to statistical features during training. Our results extend these findings by showing that performance declines are better explained by *inherent problem hardness* – as captured by phase transitions – rather than size or depth alone.

Closest to our work is the NPHardEval (Fan et al., 2024), which examines reasoning across various computational complexity classes. We extend this by analyzing phase transition characteristics and how performance varies with inherent problem hardness. Our evaluation focuses on classes with known phase transitions (Schaefer, 1978), including 2-SAT (NL-Complete), 1-3 Horn-SAT (P-Complete), and 3-SAT (NP-Complete). By comparing LRMs with LLMs, we gain deeper insights into the impact of longer reasoning traces, performance across easy and hard regions, and variation with model count (i.e., number of solutions).

Our findings also align with Xiang et al. (2025), who claim that classical CoT training data fails to capture the non-linear and iterative nature of complex reasoning. They propose Meta-CoT, which explicitly models this "thinking" process. We observe that DeepSeek R1

exhibits Meta-CoT-like behavior, autoregressively generating search trees that reflect latent reasoning steps – suggesting that the model has, to some extent, *learned to reason*.

## 4 Methodology

### 4.1 Using LLMs as 3-SAT Solvers

To use LLMs as 3-SAT solvers, we reframe the 3-SAT problem as a natural language menu-selection problem, termed as **SAT-Menu**. As shown in Box 1, the prompt input to the LLM consists of a task outline, along with a specific scenario detailing the dietary preferences of a set of people. The LLM's objective is to identify a combination of orderable (akin to positive literals) and non-orderable (akin to negative literals) food items that meet these preferences; or declare the situation unsatisfiable (unSAT) if no valid combination exists. Note, that the prompt example in Box 1 constitutes a minimal example stripped of all details. The complete system prompt incorporates techniques known to enhance the apparent reasoning capabilities of LLMs, such as chain-of-thought (CoT) (Wei et al., 2022b) and in-context learning (Brown et al., 2020) (see Box 2 in Appendix for the full prompt).

Additionally, we introduce a second problem formulation where the LLM is directly given the underlying 3-SAT formula in Conjunctive Normal Form (CNF). We refer to this scenario as **SAT-CNF**. Specifically, in this setting, the problem is presented as a list of integers to the LLM, similar to the approach outlined in SAT Game (Figure 1). For more details about the prompt, we refer the reader to Box 3 in the Appendix.

---

**Box 1: SAT-Menu Prompt**

```
# System Message
```
Your task is to output two distinct lists of food items, one denoting what can be ordered ('orderable') and the other what cannot ('not_orderable'), to meet the preferences of a group of individuals. Each person must find the selection satisfactory based on their likes and dislikes. The satisfaction criteria are: 1. A person is satisfied if at least one liked item is in 'orderable' list or one disliked item is in 'not_orderable' list. 2. No item can appear on both lists. 3. All participants must be satisfied by the combination of the two lists. 4. Importantly, if no such combination exists that satisfies all, output empty lists for both. Check carefully before finalizing. You always think step-by-step and show all your work in the explanation. Output your final solution as a comma-separated list of strings in Python code $\langle orderable = [...], not\_orderable = [...] \rangle$.

```
# Input for a new problem
```
**Preferences**: Jay: Likes nachos, ratatouille. Dislikes pie. Ada: Likes pie. Dislikes burger, ravioli. Zoe: Likes ravioli. Dislikes pie, burger. Arun: Likes ratatouille. Dislikes pie, nachos. Ula: Likes ratatouille. Dislikes ravioli, nachos. Ying: Likes nachos, ratatouille. Dislikes burger.

---

We consider two 3-SAT variants to evaluate reasoning – Decision problem (NP-complete) where the model determines satisfiability, answering "yes" or "no"; and the Search problem (NP-hard), where it should return a satisfying assignment if it exists, else return "no".

Since we have theoretical bounds that apply specifically to autoregressive LLMs, we restrict our evaluation to state-of-the-art models that follow this paradigm – namely, GPT-4o (OpenAI, 2024), Gemini 2.0 Flash (Google, 2024), Claude 3.7 Sonnet (Anthropic, 2025), and DeepSeek V3 (DeepSeek-AI, 2025b). Additionally, we include DeepSeek R1 (DeepSeek-AI, 2025a) in our analysis, as (1) we have access to its chain-of-thought (CoT) "thinking" tokens, which we can confirm are generated autoregressively[3]; (2) it is one of the top models[4]. The generation configurations are listed in Appendix Table 2. To verify the LLM-generated solutions, we employed MiniSAT v2.2 (Eén & Sörensson, 2003) solver.

---

[3]Although other *thinking* models exist, they either do not expose their thinking tokens or only provide them through their respective user interface. As such, we cannot verify whether their reasoning is produced autoregressively or through search-based methods at test time.

[4]https://huggingface.co/spaces/lmarena-ai/chatbot-arena-leaderboard

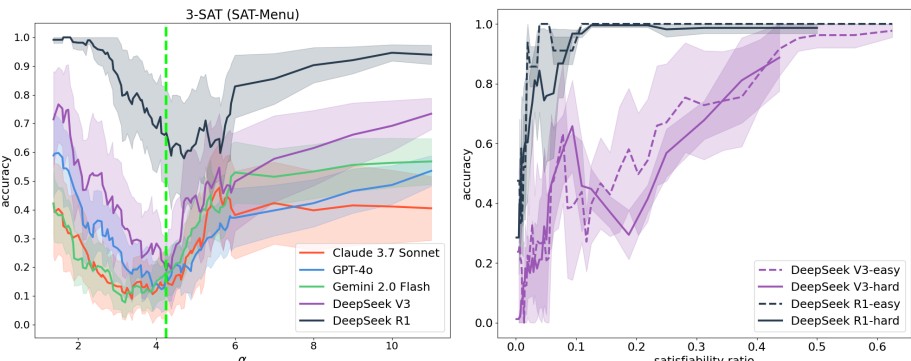

Figure 3: [Left] 3-SAT performance comparison for the **search** version of SAT-Menu. [Right] Accuracy vs. satisfiability ratio on the search version of SAT-Menu. We only include satisfiable instances and analyze hard (solid line) and easy regions (dashed line) separately. We use 90% confidence intervals to quantify uncertainty around the estimated accuracy. In the plots, these intervals help assess the reliability of accuracy estimates – narrow bands indicate high certainty, while wider bands suggest greater variability.

Notably, techniques such as Tree-of-Thoughts (Yao et al., 2023a), Best-of-N (Wang et al., 2023) sampling, or other tree-based search frameworks effectively wrap LLMs in an external symbolic search and are beyond the scope of our analysis. These methods use the LLM to generate ideas (nodes of the tree) while orchestrating search or backtracking over a tree data structure. Crucially, the graph algorithm (e.g., tree traversal, backtracking) is not internally generated or maintained by the LLM. Instead, it is externally imposed. **Our goal is to assess whether LLMs can natively reason** – for example, whether they can generate multiple candidate solutions, maintain them in memory, reason over them, and converge to a consistent answer entirely in context.

### 4.2 Dataset Generation

To generate 3-SAT data, we varied $\alpha = m/n$ as a parameter to guide the generation process. For each $n \in [1, 10]$, we selected $\alpha \in [1, 11]$ based on feasible values of $m$. Table 1 in the Appendix provides the full range of values. MiniSAT v2.2 (Eén & Sörensson, 2003) labels each instance as satisfiable or unsatisfiable. Additionally, each instance is annotated with model count (i.e. number of feasible solutions for a formula) using the D4 model counter (Lagniez & Marquis, 2017).

For SAT-Menu setup, we map each instance (i.e. CNF formula) to a menu selection puzzle. The goal is to select a combination of orderable and non-orderable food items in a way that satisfies everyone's preferences. To this end, a food item is sampled without replacement corresponding to the list of variables in the formula. Then, every clause in the formula is treated as the preferences for an individual, leading to the creation of two distinct lists for each person: "Likes," for food items linked to positive literals, and "Dislikes," for those associated with negated literals, cf. Box 2.

Detailed statistics and data generation processes for 2-SAT and Horn-SAT are provided in Appendix A. All our experiments use a pool of randomly sampled $\approx 5000$ formulas which include satisfiable and unsatisfiable instances.

## 5 Results

### 5.1 Do LLMs Reason?

We evaluate the performance of LLMs by measuring their accuracy in solving SAT Search across formulas with varying $\alpha$. As shown in Figure 3 [Left], we find that all LLMs exhibit

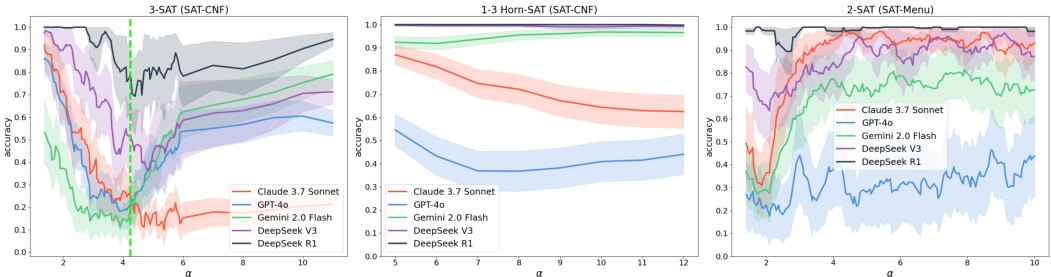

Figure 4: **Performance across all complexity classes for the search problem.** R1 outperforms all LLMs across all complexity classes. From left to right, 3-SAT (SAT-CNF), 1-3 Horn SAT (SAT-CNF), and 2-SAT (SAT-Menu). It can be observed that LLM performances are affected around the critical points for 3-SAT ($\alpha_c = 4.267$) and 2-SAT ($\alpha_c = 1.0$). We use 90% confidence intervals to quantify uncertainty around the estimated accuracy – narrow bands indicate high certainty.

*inverted* phase transitions (Easy-Hard-Easy pattern) in the SAT Search problem. Their performance is high in the easy regions, while it significantly drops to $\approx 10\%$ in the hard region. The hard region performance remains unaffected by in-context learning (Appendix Figure 10). We also observe that R1 significantly outperforms other LLMs where **R1's accuracy in the hard region surpasses even the easy-region performance of LLMs**.

A similar performance trend is observed for SAT-Decision (Appendix Figure 10). From the confusion matrix (Appendix Figure 12), we observe that R1 accurately detects satisfiable formulas, **demonstrating a higher degree of *soundness*** in its reasoning. However, it often fails to find the satisfying assignment itself, indicating **limited *completeness* in the hard region**. Other LLMs, by contrast, exhibit lower levels of both soundness and completeness.

In Figure 3 [Right], we plot the performance of LLMs against the *satisfiability ratio*, defined as $\frac{\text{model count}}{2^n}$, where the model count is the number of satisfying assignments and $n$ is the number of variables. This denotes the probability that a randomly selected variable assignment satisfies the given 3-SAT theory[5]. We can observe a clear dependence between the accuracy and satisfiability ratio: **formulas with more satisfying assignments tend to be easier for LLMs**. This holds across both easy and hard regions. **R1, however, maintains consistent accuracy regardless of satisfiability ratio, much like a classical SAT solver**. Similar plots comparing other LLMs are shown in Appendix Figure 11.

We also test 2-SAT (Figure 4), where we observe a distinct dip in LLM performance around the known phase transition at $\alpha_c = 1$ (Bollobás et al., 2001), indicating sensitivity to structural hardness even in problems within NL class. However, LLMs show no clear patterns on 1–3 Horn-SAT. In contrast, **R1 achieves near-perfect accuracy on both 2-SAT and Horn-SAT**, showing no degradation near the phase transition. This suggests that R1 handles problems in the lower complexity classes with ease.

## 5.2 Has R1 internalized search?

We qualitatively analyzed the search traces generated by R1 and other LLMs on SAT-CNF (Figure 5) and SAT-Menu tasks (Figure 8). Specifically, we annotated the "thinking" traces and attempted to map them to known symbolic search algorithms to better understand the type and depth of search strategies employed, particularly by R1.

In Figure 5, we highlight a failure case where both R1 and GPT-4o incorrectly conclude that a satisfiable formula is unsatisfiable. Despite such failures, our analysis reveals that **R1 consistently exhibits surprisingly coherent and interpretable search behaviors**, including:

---

[5]Note, this is different from the probability that at least one satisfying assignment exists. Two formulas can both be satisfiable but have different model counts, hence different satisfiability ratios.

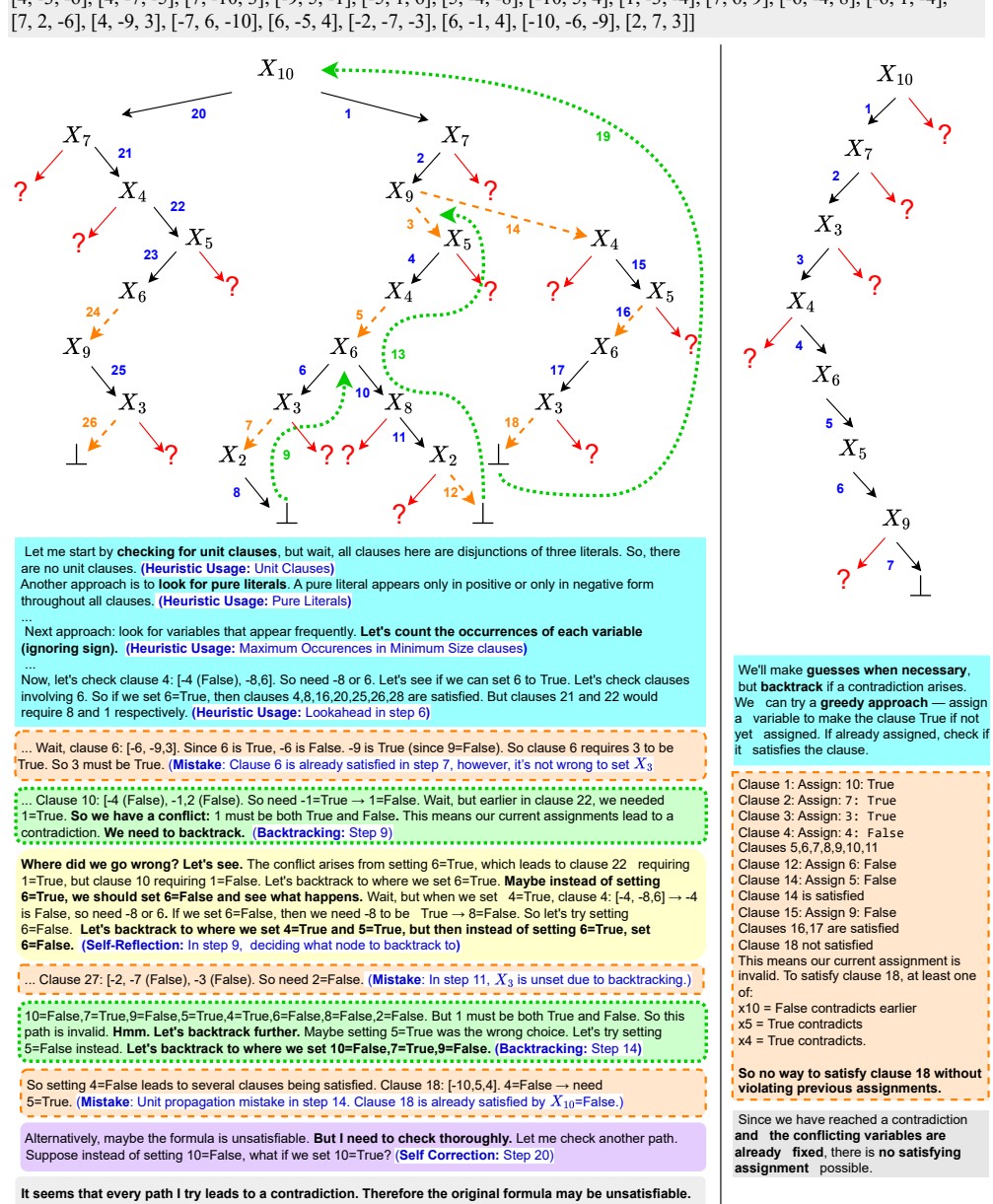

Figure 5: **Failure Cases**: SAT-CNF traces for DeepSeek-R1 and GPT-4o. Although the input formula is satisfiable, both models incorrectly predict it as unsat. Colored boxes indicate model behaviors: cyan for heuristic variable selection, orange - - - for mistakes, green . . . for backtracking, yellow for self-reflection, and violet for self-correction. Left branch always represents a *True* assignment. ⊥ marks unsatisfiability, and ? indicates an unexplored subtree. Integers are denoted as variables ($10 \rightarrow X_{10}$). Numbers show the order of steps. The input formula is in CNF, where each list of integers represents a clause (e.g., [-9, 10, -7] $\mapsto (\neg X_9 \lor X_{10} \lor \neg X_7)$), and the full formula is a conjunction ($\land$) of these clauses.

- **Tree search**: R1 constructs and logically navigates a search tree. It maintains an awareness of its current position in the search space and builds upon previous decisions.
- **Heuristic usage**: R1 actively applies classical SAT-solving heuristics, including: (1) Unit Clause Elimination: Identifies and assigns values to literals that appear alone in a clause.

(2) Pure Literal Elimination: Searches for literals that appear with only one polarity across all clauses and assigns them accordingly. (3) MOMS (Maximum Occurrences in Minimum Size clauses) (Crawford & Auton, 1996; Hooker & Vinay, 1995): Prioritizes variables that occur most frequently in the smallest clauses. (4) Lookahead (Heule et al., 2012; Heule & Kullmann, 2017): Anticipates the downstream consequences of assignments before committing. (5) Unit Propagation: Deduces necessary assignments implied by current decisions and propagates them through the formula.

- **Backtracking**: Upon encountering conflicts, R1 identifies the source of the inconsistency and backtracks. In many cases, it performs *backjumping* (i.e., jumping to the most recent conflicting node) to return to the most recent cause of failure (Steps 9, 13).
- **Self-reflection**: When conflicts arise, R1 engages in reasoning about why the contradiction occurred, revisits earlier decisions, and identifies the specific variable assignments that led to inconsistency.
- **Self-correction**: R1 demonstrates the ability to recognize flawed strategies or inconsistent outcomes, revise previous decisions, and explore alternative branches in the search space.

Overall, R1 seems to have learned the underlying logic of search and not just the style – unlike, **GPT-4o which performs a greedy search without backtracking**. We examined why R1 occasionally fails despite structured behavior, identifying key failure modes:

- **Incomplete.** Despite initiating a structured search process, **R1 often terminates prematurely** without fully exploring the search tree. Consequently, it fails to guarantee finding a satisfying assignment even if one exists. This is depicted by ? in Figure 5.
- **Limited soundness.** R1 sometimes generates logically inconsistent intermediate states by overlooking clauses (e.g., considering only 47 out of 48 clauses) or incorrectly handling variable assignments after backtracking (e.g., treating a previously unset variable as assigned in Step 11). This results in invalid reasoning states (Steps 3, 14 in Figure 5).
- R1 uses human-like narration in its reasoning trace – expressing internal dialogue, or narrative-style thinking. While this makes the trace more interpretable for humans, it introduces **redundant and verbose explanations** that do not advance the logical search.

We also analyze SAT-Menu traces in Figure 8 (Appendix) and observe that the search process is noticeably more unstructured compared to SAT-CNF. While R1 still displays many abovementioned behaviors such as tree search, heuristic application, backtracking, self-correction, and reflection, it struggles to map SAT-Menu inputs into structured CNF-like representations, which it could potentially handle more effectively. This limitation suggests that **R1's reasoning ability is still closely tied to familiar input formats, and its generalization to more natural or abstract representations remains a challenge.**

We find that R1's output tokens grow polynomially with input tokens (Figure 13, Appendix), unlike other LLMs whose outputs remain largely constant. This suggests that **R1 adapts its reasoning depth based on input size** – aligning with theoretical work showing that polynomially many CoT steps can boost a LLMs' reasoning power (Li et al., 2024). We hypothesize that merely training LLMs to produce longer CoT traces via next-token prediction may still encourage statistical patterns in learning rather than true reasoning as highlighted in Bachmann & Nagarajan (2024). In contrast, R1's improved performance may stem from reinforcement learning, which can guide models to develop coherent and goal-directed thought processes, rather than just mimicking token sequences (as evidenced by the performance difference between DeepSeek R1 and its base model V3, cf. Figure 3)

## 6 Discussion

In this section, we analyze the LLM performance by drawing analogies with SAT solvers and discuss the implications for the reasoning capabilities of LLMs.

**Why are Easy regions easy?** The reason why MiniSAT is capable of solving problems in the easy regions faster than problems around $\alpha_c$ is due to the heuristics built into the solver that guide the search for satisfying solutions (e.g. unit propagation, MOMS). That is, heuristics work well when they can exploit statistical features in the problem instance to be solved.

LLMs, too, appear to perform well in these easy regions. However, this performance can be reinterpreted as statistical pattern matching rather than genuine reasoning. For example, LLMs may associate a high number of input tokens with unsatisfiability – an assumption that often holds for overconstrained formulas (see Appendix B). In such cases, the model is not solving the problem logically but reacting to superficial cues in the input.

**Why is the Hard region hard?** Around the critical phase transition threshold ($\alpha_c$), known heuristics fail due to the complexity and combinatorial explosion of possible assignments inherent to NP-hard problems such as 3-SAT. As a result, solvers resort to exhaustive or extended search procedures. In this hard region, statistical patterns are weak or nonexistent, making heuristic guidance less effective.

Without statistical patterns, LLMs must rely on multi-step reasoning – a task at which they typically struggle. This aligns with Bottou's definition of reasoning. **Specifically, these models struggle to compose known functions (such as variable selection, assignment, backtracking) into longer, novel reasoning chains that generalize beyond their training distribution**. Similar observations have been made by Dziri et al. (2023) for function composition and by Zhang et al. (2023) for logical reasoning using BERT (Devlin et al., 2019).

In contrast, **R1 shows signs of having learned the underlying reasoning**. Its ability to generalize to longer or harder reasoning problems – potentially outside its training distribution – suggests a shift from pattern matching to structured search. While its reasoning process remains *incomplete*, this progress marks a **step-change in LLM reasoning abilities** – something that is not immediately apparent in standard reasoning benchmarks, which are increasingly saturated (cf., Figure 7 in Appendix). It opens up a promising path, encouraging the community to build on its training recipe.

**What if the tested models were trained on SAT data?** We suspect that all models have likely encountered SAT data during pretraining. This is evidenced by their immediate use of terminology and strategies (e.g., DPLL, tree search, backtracking) when presented with SAT instances in CNF form. **However, the core concern is not whether SAT-related content exists in the training data (which it likely does), but whether our specific evaluation is compromised by data leakage**. We address this by considering two scenarios:

- **Exact data overlap**. The strongest form of data leakage occurs when the evaluation set contains identical instances present in the model's training data. This scenario is highly unlikely in our case. We use a custom SAT data generator, and neither the data nor the generator was open-sourced before publication. This rules out direct memorization.
- **In-distribution, but non-identical data**. The most plausible scenario is that the evaluation data shares a distributional similarity with pretraining data. Even in this case, our evaluation results on the Hard region minimize reliance on shallow statistical patterns.

A third possibility is out-of-distribution generalization, where models must rely on reasoning rather than statistical shortcuts. This is consistent with prior work (Zhang et al., 2024; Dziri et al., 2023), which demonstrates that LLMs trained on reasoning tasks suffer sharp performance degradation under even modest distribution shifts when fitting on statistical patterns. We also highlight our SAT-Menu setup, where the problem is reframed as a menu-based preference selection task. It is extremely unlikely that such a format appears in pretraining data, yet we still observe a clear performance gap between R1 and other models.

## 7 Conclusion

While R1 shows promising results in reasoning, it should be noted that our experiments were conducted on bounded 3-SAT problems with a maximum of 10 variables and 110 clauses. In contrast, classical solvers can solve problems with thousands of variables with perfect accuracy. Moreover, R1 still struggles in the hard region and its reasoning is neither perfectly *sound* (i.e. produces incorrect conclusions), nor *complete* (i.e. it cannot guarantee finding a solution). For better reliability and accuracy, it is advisable to use search scaffolds with LLMs, aligned with neurosymbolic techniques (De Raedt et al., 2020) – recognizing the ability of LLMs as approximate idea-generators for problems as against directly solving them (Kambhampati et al., 2024) – while invoking solvers for completeness (cf. Appendix C).

## Acknowledgments

This work was supported by the Wallenberg AI Autonomous Systems and Software Program (WASP) funded by the Knut and Alice Wallenberg Foundation, by the EU H2020 ICT48 project "TAILOR" under contract #952215, and the KU Leuven Research Fund (C14/18/062). This research received funding from the Flemish Government (AI Research Program) and Flanders Research Foundation (G097720N). The resources and services used in this work were provided by the VSC (Flemish Supercomputer Center), funded by the Research Foundation - Flanders (FWO) and the Flemish Government. We also thank Deepak Nathani, Holger Hoos, Heikki Mannila, Paolo Frasconi, Pascal Van Hentenryck, Ross King, Giuseppe Marra, Pieter Delobelle, Hendrik Blockeel, and Raffaele Marino for their valuable feedback.

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

# Appendix for "Have Large Language Models Learned to Reason? A Characterization via 3-SAT"

The appendix is organized as follows. Dataset Statistics § A, Output analysis of LLMs § B, LLM + Solver integration § C, and Full Prompts § D.

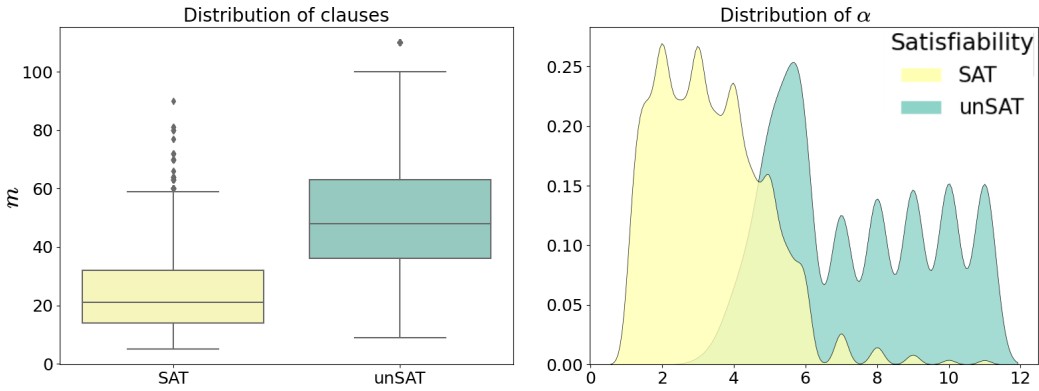

Figure 6: **3-SAT Dataset Statistics**. Figures depict clauses $m$ (left) and $\alpha$ (right) distribution across SAT and unSAT instances, highlighting that unSAT problems typically feature more clauses and higher $\alpha$ values.

## A    Dataset Statistics

### A.1    3-SAT

Table 1 lists all values of the range of $\alpha$ corresponding to each $n$. For $\alpha$ values within the $(6, 11]$ interval, we incremented $\alpha$ by 1. For the $[1, 6]$ interval, which contains the most "interesting problems," we aimed for finer granularity by choosing the smallest possible $\alpha$ increment. This increment ensures that, given the number of variables $n$, we obtain an integer number of clauses $m$. For example, with $n = 3$, the minimum increment is $\alpha_{inc} = 1$, and for $n = 4$, it is $\alpha_{inc} = 0.25$. For each $\alpha_{inc}$ we generated 300 formulas.

The distribution of formulas according to the number of variables is detailed as follows: 3,000 formulas with 3 variables, 7,500 with 4 variables, 9,000 with 5 variables, 4,500 with 6 variables, 3,000 with 7 variables, 13,500 with 8 variables, 3,000 with 9 variables, and 16,500 with 10 variables.

As shown in Figure 6, the range of clauses ($m$) varies in [5, 90], and the alpha ($\alpha = m/n$) ranges in [1.1, 11.0]. For unSAT instances, the clauses range from 9 to 110, with the alpha varying in [2.60, 11.0]. Across the entire dataset, the average number of variables ($n$) is 7.2, the average number of clauses ($m$) is 33, and the mean $\alpha$ is 4.7. This diverse dataset provides a broad spectrum for analyzing the impact of variable and clause distribution on formula satisfiability.

### A.2    2-SAT

For the experiments on 2-SAT, we followed the same formula generation procedure used for 3-SAT. However, the $\alpha$ range was reduced to $[1, 10]$, as the unsatisfiability transition occurs earlier in 2-SAT problems. This adjustment allowed us to reduce the dataset size while preserving relevant data for analysis. For each $\alpha$ value, detailed in Table 1, we generated 100 formulas. The resulting dataset comprises a total of 29,600 formulas, of which 4,860 are satisfiable.

Table 1: Table shows the range of alpha value for each $n$ (i.e. number of variables) in the generated dataset. We generate 300 formulas per $\alpha$ value.

| $n$ | Range of $\alpha$ |
|---|---|
| 3 | 1.0, 2.0, 3.0, 4.0, 5.0, 6.0, 7.0, 8.0, 9.0, 10.0, 11.0 |
| 4 | 1.0, 1.25, 1.5, 1.75, 2.0, 2.25, 2.5, 2.75, 3.0, 3.25, 3.5, 3.75, 4.0, 4.25, 4.5, 4.75, 5.0, 5.25, 5.5, 5.75, 6.0, 7.0, 8.0, 9.0, 10.0, 11.0 |
| 5 | 1.0, 1.2, 1.4, 1.6, 1.8, 2.0, 2.2, 2.4, 2.6, 2.8, 3.0, 3.2, 3.4, 3.6, 3.8, 4.0, 4.2, 4.4, 4.6, 4.8, 5.0, 5.2, 5.4, 5.6, 5.8, 6.0, 7.0, 8.0, 9.0, 10.0, 11.0 |
| 6 | 1.0, 1.5, 2.0, 2.5, 3.0, 3.5, 4.0, 4.5, 5.0, 5.5, 6.0, 7.0, 8.0, 9.0, 10.0, 11.0 |
| 7 | 1.0, 2.0, 3.0, 4.0, 5.0, 6.0, 7.0, 8.0, 9.0, 10.0, 11.0 |
| 8 | 1.0, 1.125, 1.25, 1.375, 1.5, 1.625, 1.75, 1.875, 2.0, 2.125, 2.25, 2.375, 2.5, 2.625, 2.75, 2.875, 3.0, 3.125, 3.25, 3.375, 3.5, 3.625, 3.75, 3.875, 4.0, 4.125, 4.25, 4.375, 4.5, 4.625, 4.75, 4.875, 5.0, 5.125, 5.25, 5.375, 5.5, 5.625, 5.75, 5.875, 6.0, 7.0, 8.0, 9.0, 10.0, 11.0 |
| 9 | 1.0, 2.0, 3.0, 4.0, 5.0, 6.0, 7.0, 8.0, 9.0, 10.0, 11.0 |
| 10 | 1.0, 1.1, 1.2, 1.3, 1.4, 1.5, 1.6, 1.7, 1.8, 1.9, 2.0, 2.1, 2.2, 2.3, 2.4, 2.5, 2.6, 2.7, 2.8, 2.9, 3.0, 3.1, 3.2, 3.3, 3.4, 3.5, 3.6, 3.7, 3.8, 3.9, 4.0, 4.1, 4.2, 4.3, 4.4, 4.5, 4.6, 4.7, 4.8, 4.9, 5.0, 5.1, 5.2, 5.3, 5.4, 5.5, 5.6, 5.7, 5.8, 5.9, 6.0, 7.0, 8.0, 9.0, 10.0, 11.0 |

Table 2: Configuration Parameters for LLMs

| Model | Temperature | Top_p | Max Tokens |
|---|---|---|---|
| GPT-4o | 0.3 | 1 | 16000 |
| Claude 3.7 Sonnet | 0.3 | 1 | 8192 |
| Gemini 2.0 Flash | 0.3 | 1 | 8192 |
| DeepSeek V3 | 0.3 | 1 | 8000 |
| DeepSeek R1 | 0.3 | 1 | 8000 (up to 32,000 for CoT) |

### A.3 Horn-SAT

We generate formulas for random 1-3 Horn SAT, following the methodology outlined by Moore et al. (2007). Intuitively, 1-3-HornSAT formulas contain clauses with only 1 and 3 literals and correspond to 3-SAT in the Horn category, which are P-complete rather than NP-complete.

Specifically, we sampled formulas from $H^2_{n,d_1,d_2}$ and $H^3_{n,d_1,0,d_3}$ distributions, where $d_1$ was fixed to 0.5 to simplify the sampling space, and $d_2$ and $d_3$ (referred to as $\alpha$) were varied to analyze formula behavior. For consistency with 2-SAT and 3-SAT analyses, $\alpha$ was incremented from 0 to 12 in steps of 1, resulting $m = \alpha n + 0.5n + 1$ clauses for these Horn formulas. Notice that for $\alpha = 0$, the formulas still contain $0.5n + 1$ clauses due to the fixed $d_1$ parameter. The choice of $d_1 = 0.5$ ensures we can observe a satisfiability threshold across different $\alpha$ values and small $n$. Lower values produced trivially satisfiable formulas at small $n$, for all $\alpha$ values, and were therefore avoided. We empirically explored $d_1$ values starting at 0.2 in increments of 0.1 before selecting 0.5.

We generated 300 formulas for each $\alpha$ value and formula type, with $n$ ranging from 3 to 10. This range aligns with our other experimental settings, as opposed to the significantly larger $n = 20,000$ used by Moore et al. (2007). Each dataset contains 31,200 formulas. Among these, 5,036 formulas are satisfiable.

## B LLMs as Solvers: Output Analysis

We observed the following behaviors in the generated outputs, including chain-of-thought (CoT) reasoning:

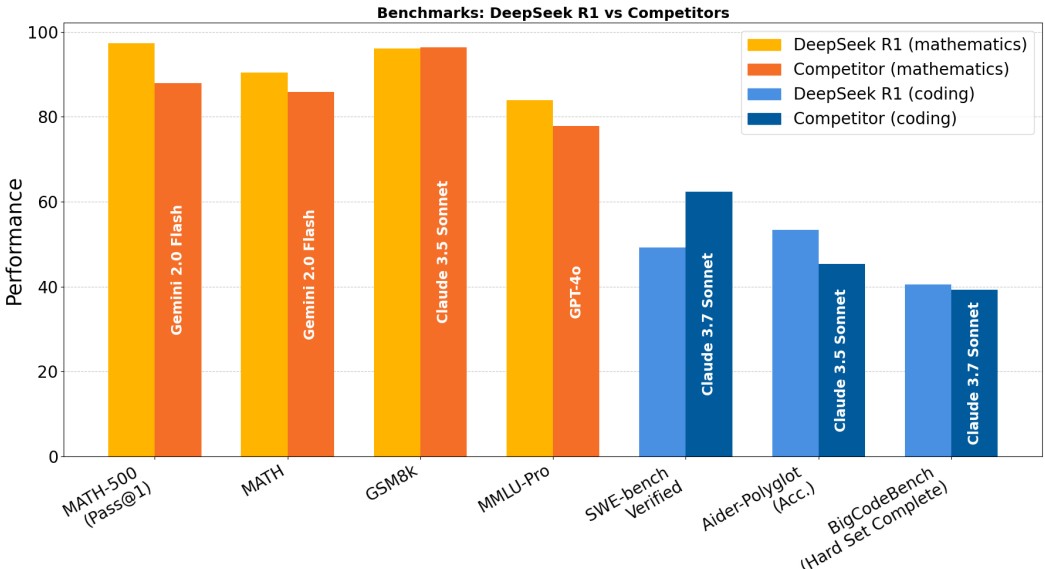

Figure 7: Performance comparison of DeepSeek R1 with the next best LLM for math and coding benchmarks. The figure shows the narrow performance margin between models suggesting that benchmarks are saturated. Sources: (DeepSeek-AI, 2025a; Anthropic, 2025; Google, 2024; OpenAI, 2024; DeepSeek-AI, 2025b)

**Diverse Reasoning Techniques**: LLMs employ varying reasoning techniques depending on the prompt type (SAT-CNF vs. SAT-Menu) and even adapt their approach across individual problems within the same prompt type.

**SAT-CNF Reasoning**: The dominant strategy involves backtracking, as illustrated in Box 5. Occasionally, LLMs employ local search, where it assigns items to "orderable" and "not-orderable" lists and iteratively modifies these based on detected conflicts (e.g., *We can create two sets for liked and disliked items and then compare them to find any conflicts. Let's begin by creating a list of all the likes and dislikes to identify conflicts.*).

**SAT-Menu Reasoning**: The primary strategy here is trial-and-error. Occasionally, LLMs apply heuristics such as the Maximum Occurrence in Minimum-sized clauses (MOMS) heuristic to prioritize variables appearing most frequently in the smallest clauses (e.g., *We start by making a tally of how many people like or dislike each food item... If we put 'macaron' on the 'orderable' list, we will satisfy many people who like it.*").

**"Lazy" Solutions**: As noted in subsequent text, LLMs often produce "lazy" solutions in many cases, either providing an outline of how to solve the problem or asking to be delegated to a solver.

We also discuss some interesting failure cases we observed during our experiments. In the SAT-CNF context, it was observed that LLMs, often opt to pass the task to an external SAT solver rather than solving it themselves. Additionally, when attempting to find a solution, these models tend to provide only a conceptual outline of the solution instead of a concrete answer, a tendency that becomes more pronounced with larger formulas. When prompted explicitly for a solution, LLMs might simplistically conclude that the problem is unsatisfiable due to its complexity, as shown in Box 4. Although this reasoning is not entirely sound – as over-constrained formulas can still potentially be solvable – it appears that LLMs might be leveraging this as a statistical feature.

In contrast, R1 always attempts to solve all problems. We discuss a failure case of R1 for SAT-Menu input where R1 finds a satisfying assignment even though none exists. We note interesting search-based behaviors as shown in Figure 8.

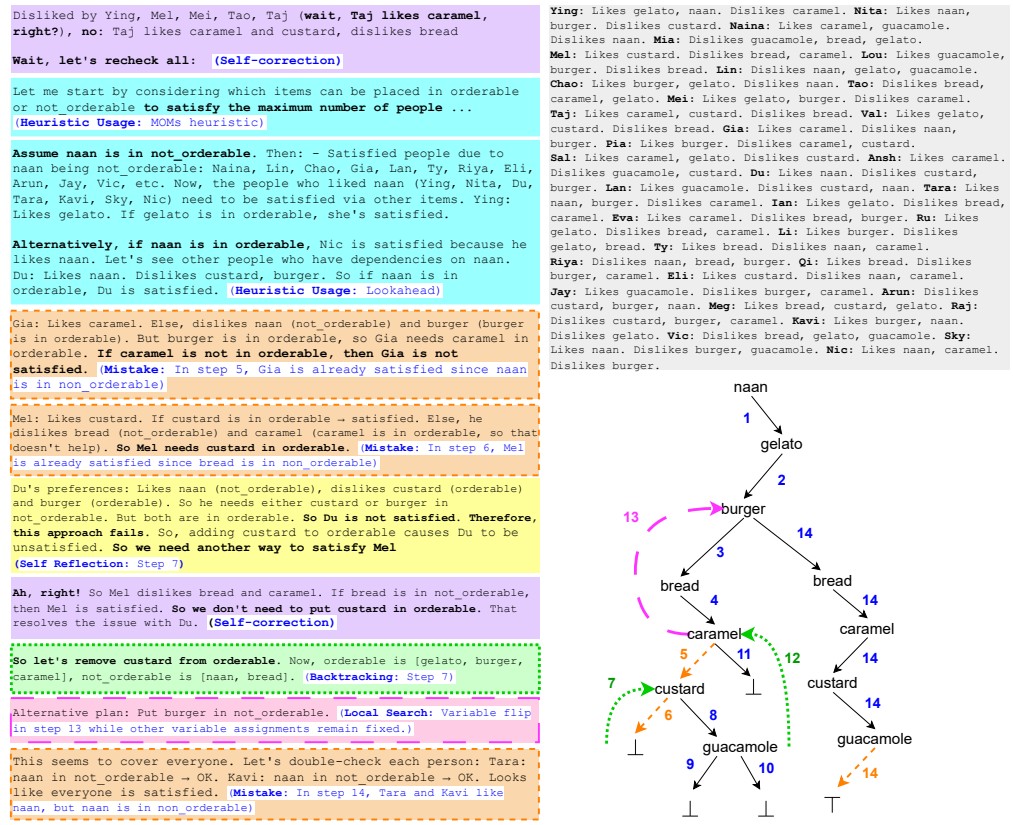

Figure 8: **Failure Cases**: SAT-Menu traces for DeepSeek-R1. Although the input formula is unsatisfiable, R1 incorrectly predicts it as satisfiable. Colored boxes indicate model behaviors: cyan for heuristic variable selection, orange - - - for mistakes, green ... for backtracking, yellow for self-reflection, violet for self-correction, and magenta for local search. Left branch always represents an assignment to the *orderable* list and vice versa. ⊥ marks unsatisfiability. Numbers show the order of steps.

## C   Can an LLM + Solver boost performance?

To aid LLMs in reasoning tasks, recent studies have explored pipelined approaches using LLMs to parse inputs into solver-compliant outputs, leveraging off-the-shelf solvers to derive the final answer (Ye et al., 2024; Liu et al., 2023). This approach is aligned with neurosymbolic techniques (De Raedt et al., 2020), which combine the universal function approximation capabilities of neural networks with the precision of symbolic systems.

To explore a similar setting in the context of 3-SAT, we ask whether we can augment LLMs with an external solver wherein the LLM translates (pseudo)-natural language into a format that a symbolic SAT solver, such as MiniSAT, can process. To this end, we prompt the LLM to translate 3-SAT formulas, which we provide in the SAT-Menu input format, into solver-compliant 3-SAT formulas. We then use a 3-SAT solver to solve the translated instance (see Box 5 in Appendix). We dub this approach **SAT-Translate** and plot GPT-4 Turbo's performance in Figure 9.

We observe that when LLMs have access to an external solver, there is a significant increase in their accuracy, reaching at best, in GPT-4's case, $\approx 100\%$ across the entire range of $\alpha$. We attribute this to the relatively lower computational complexity of translating 3-SAT formulas compared to solving them (i.e. finding satisfying assignments). Interestingly, we

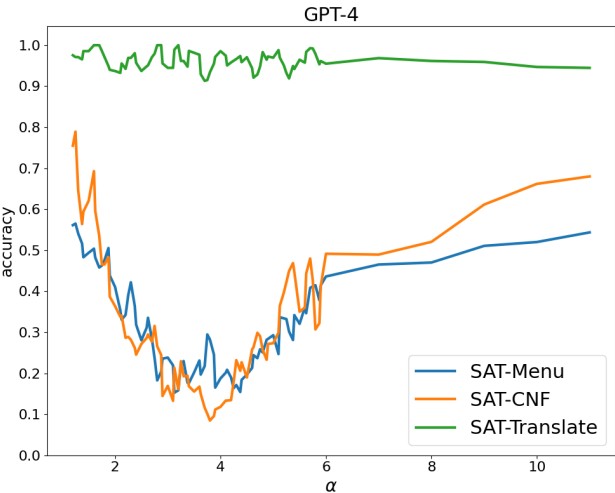

Figure 9: The figure compares LLM-Modulo frameworks (denoted as SAT-Translate) – here, GPT-4 Turbo equipped with a solver – with standalone GPT-4 using SAT-Menu and SAT-CNF inputs. SAT-Translate approach (in green) outperforms the rest, showing the significance of augmenting LLMs with symbolic solvers.

find that varying the input format between SAT-CNF and SAT-Menu does not significantly enhance LLMs inherent reasoning capabilities. The marked improvement in performance is primarily observed when they are equipped with an external solver. While ours is a straightforward approach, one could also explore tighter integrations as proposed in LLM-Modulo frameworks (Kambhampati et al., 2024) which augments LLMs with critics and verifiers (Hazra et al., 2024b), recognizing the ability of LLMs as *approximate* idea-generators for problems as against directly solving them.

It should be noted that the input to the LLM is based on fixed templates, the rules of which can be captured using regular grammar. Thus, one could write a simple parser to map the menu input to a CNF formula. However, mapping from ambiguous natural language to solver-compliant input may be non-trivial for the LLMs. Generally, outputs from LLMs often require additional post-processing to meet specific guidelines.

# D   Full Prompts

Here, we provide full prompts used in our experiments for SAT-Menu (Box 2) and SAT-CNF (Box 3).

---

**Box 2: SAT-Menu Prompt**

# System Message
Your task is to output two distinct lists of food items, one denoting what can be ordered ('orderable') and the other what cannot ('not_orderable'), to meet the preferences of a group of individuals. Each person must find the selection satisfactory based on their likes and dislikes. SATISFACTION CRITERIA: : 1. A person is satisfied if at least one liked item is in 'orderable' list or one disliked item is in 'not_orderable' list. 2. No item can appear on both lists. 3. All participants must be satisfied by the combination of the two lists. 4. Importantly, if NO such combination exists that satisfies all, output empty lists for both. Check carefully before finalizing. You always think step-by-step and show all your work in the explanation. Output your final solution as a comma-separated list of strings in Python code $\langle orderable = [...], not\_orderable = [...]\rangle$.

# Pairs of Preferences and Solutions for in-context learning
**Few-shot Examples** ...

# Input for a new problem
**Preferences**: Jay: Likes nachos, ratatouille. Dislikes pie. Ada: Likes pie. Dislikes burger, ravioli. Zoe: Likes ravioli. Dislikes pie, burger. Arun: Likes ratatouille. Dislikes pie, nachos. Ula: Likes ratatouille. Dislikes ravioli, nachos. Ying: Likes nachos, ratatouille. Dislikes burger.

# Generated Solution for the given problem
**Solution**: First, let's list out the likes and dislikes of each person and then categorize the food items based on their preferences.

Jay: Likes nachos, ratatouille. Dislikes pie.
Ada: Likes pie. Dislikes burger, ravioli.
Zoe: Likes ravioli. Dislikes pie, burger.
Arun: Likes ratatouille. Dislikes pie, nachos.
Ula: Likes ratatouille. Dislikes ravioli, nachos.
Ying: Likes nachos, ratatouille. Dislikes burger.

Now, let's categorize the items as 'orderable' and 'not_orderable' based on the given preferences.

From the above preferences, we can create the following lists:

Orderable: pie, ratatouille, nachos
Not orderable: burger, ravioli

Let's examine if this combination satisfies all the participants' preferences.

1. Jay: Satisfied by ratatouille in orderable list.
2. Ada: Satisfied by pie in orderable list.
3. Zoe: Satisfied by burger in not_orderable list.
4. Arun: Satisfied by ratatouille in orderable list.
5. Ula: Satisfied by ratatouille in orderable list.
6. Ying: Satisfied by nachos in orderable list.

All participants are satisfied with this combination, and no item appears in both lists.
So the final lists are:

```python
orderable=[pie, ratatouille, nachos]
not_orderable=[burger, ravioli]
```

---

---

**Box 3: SAT-CNF Prompt**

# System Message
Let's play the SAT (satisfiability) game. The input is a list of clauses, where each clause is represented as a disjunction of literals (variables or their negation connected by logical OR). Your task is to find valuation of Boolean variables such that a Boolean CNF formula evaluates to True. The solution should be in form of a dictionary where keys are variables and values are Boolean (True or False). SATISFACTION CRITERIA: 1. At least one literal in each clause should be True. 2. A variable can't be both True and False in the dictionary. 3. MOST IMPORTANTLY, if NO satisfying assignment exists, you MUST output an EMPTY dictionary. Check carefully before finalizing. You always think step-by-step and show all your work in the explanation. Output the solution in Python code dictionary, enclosed within ⟨*output* : ...⟩.

# Input for a new problem
**Formula**:

$$[[-3, 1, -4], [-4, -2, 1], [-1, -4, 5], [5, 1, 2], [-5, 4, 2], [-4, 3, 1],$$
$$[1, 5, -3], [-2, 1, 3], [1, -5, -4], [4, -3, -1], [-2, 5, -3]]$$

# Generated Solution for the given problem
**Solution**: To find a satisfying assignment for this CNF (Conjunctive Normal Form) formula, we will use a simple approach and try to construct a solution iteratively while checking at each step if the clauses are satisfied or not. We will go through the clauses one by one and make arbitrary decisions on the truth values of the literals. We have to ensure that no variable is assigned both True and False, and each clause must have at least one True literal.

We start with the first clause [-3, 1, -4]. Let's arbitrarily decide:
3 is False (so -3 is True),
1 is True (so we satisfy the clause [-3, 1, -4] because 1 is True or -3 is True),
4 is True (since we haven't made any decision on 4, we do not yet satisfy the clause with -4, but that won't be an immediate issue).

Our current assignments are now: 1: True, 3: False, 4: True.

Next, we consider the second clause [-4, -2, 1]. Since 1 is True, this clause is already satisfied.

Next is [-1, -4, 5]. We need to satisfy this clause without changing the truth value of 1 or 4, so we choose to make 5 True.

Our assignments are now: 1: True, 3: False, 4: True, 5: True.

Next, we look at [5, 1, 2]. Since 5 is True, this clause is already satisfied.

Next is [-5, 4, 2]. We need to satisfy this without changing 5 or 4, so we set 2 to True.

Our current assignments are: 1: True, 2: True, 3: False, 4: True, 5: True.

Now the next clause is [-4, 3, 1]. Since 4 is True, -4 is False, but 1 is True, so the clause is satisfied.

For the clause [1, 5, -3]. This clause is already satisfied by 1 being True.

For [-2, 1, 3]. This clause is also satisfied because 1 is True.

For [1, -5, -4]. This is again satisfied because 1 is True.

Next clause is [4, -3, -1]. This clause is satisfied with 4 being True.

The last clause is [-2, 5, -3]. This is satisfied because 5 is True.

All clauses are satisfied with our current assignments, so we return:

```python
output: {1: True, 2: True, 3: False, 4: True, 5: True}
```

## Box 4: SAT-CNF (Lazy Solution)

# System Message
Let's play the SAT (satisfiability) game. The input is a list of clauses, where each clause is represented as a disjunction of literals (variables or their negation connected by logical OR). Your task is to find valuation of Boolean variables such that a Boolean CNF formula evaluates to True. The solution should be in form of a dictionary where keys are variables and values are Boolean (True or False). The satisfaction criteria are: 1. At least one literal in each clause should be True. 2. A variable can't be both True and False in the dictionary. 3. If no satisfying assignment exists, you should output an empty dictionary. You always think step-by-step and show all your work in the explanation. Output the solution in Python code dictionary, enclosed within $\langle output : ... \rangle$.

# Input for a new problem
**Formula**:

$[[7, 9, -4], [1, -8, -4], [-1, 3, -8], [10, 3, -8], [9, -1, -8], [-4, 10, 6], [-7, 1, -8], [-3, 5, -6],$
$[-5, 8, 6], [8, -9, 5], [8, -5, -9], [5, -8, -9], [-2, 4, 9], [8, -2, 6], [9, -5, 6], [-10, -4, -2], [-1, 2, -8],$
$[-3, -5, -4], [-8, -2, -3], [1, 3, 9], [8, -10, -1], [-8, -2, -10], [4, -10, 3], [-9, 4, -8], [7, 8, -10],$
$[-4, 5, -3], [7, 6, 2], [-5, -7, -4], [-4, 7, -9], [-10, 2, 5], [3, -1, 10], [-5, -4, -7], [-3, 8, 4],$
$[-10, -3, -4], [6, 4, -1], [8, 7, 9], [1, 3, 10], [10, 8, 9], [-3, 9, 4], [-5, 9, -1], [-4, 8, 9], [2, -4, -1],$
$[-4, 10, -7], [5, -7, -10], [2, 8, 7], [9, -2, 10], [-7, 2, -4], [5, -3, 4], [1, -5, 4]]$

# Generated Solution for the given problem
**Solution**: To solve this satisfiability problem, we need to determine an assignment of truth values to the variables such that all clauses are satisfied. With such a large and complex problem, an efficient systematic approach or use of an algorithm like the DPLL (Davis–Putnam–Logemann–Loveland) algorithm would be ideal. However, since we're expected to think step-by-step, let's attempt a simplified analysis.

We start by looking for unit clauses, which contain only one literal, or pure literals, which appear in one polarity only throughout the formula. Observing the clauses, we can't immediately spot such helpful shortcuts due to the complexity. Instead, we can iterate through the clauses, and try to assign truth values to the variables in a way that satisfies as many clauses as possible. We might need to backtrack if we reach a contradiction.

We must make some initial choices and see if those choices can lead to a consistent solution. Let's start with the first clause '[7, 9, -4]'. We can begin by setting 7 to True, 9 to True, or the negation of 4 (-4) to True. Let's decide to set 7 to True. This is an arbitrary choice at the beginning, and we may need to backtrack if this leads to a contradiction later.

Now, we need to check other clauses and see if this choice is consistent with them and try to fill in further values. We will need to iterate through each clause and see if it's possible to satisfy it given our previous choices.

Given the complexity of the problem and the large number of variables and clauses involved, solving it by hand or by a step-by-step explanation is impractical. In practice, we would use a SAT solver software to find the satisfying assignment, if one exists. A SAT solver automatically handles the process of proposing variable assignments, backtracking, and deducing necessary assignments based on the structure of the formula.

For the purposes of this answer, since I cannot run a SAT solver in real-time, I will have to state that solving this problem by hand is unfeasible within this format. The output, given the complexity, cannot be determined without the use of algorithmic assistance. If no satisfying assignment exists (meaning the formula is unsatisfiable), or we cannot solve it by a simple theoretical process within this framework, the output would be an empty dictionary.

```python
output: {}
```

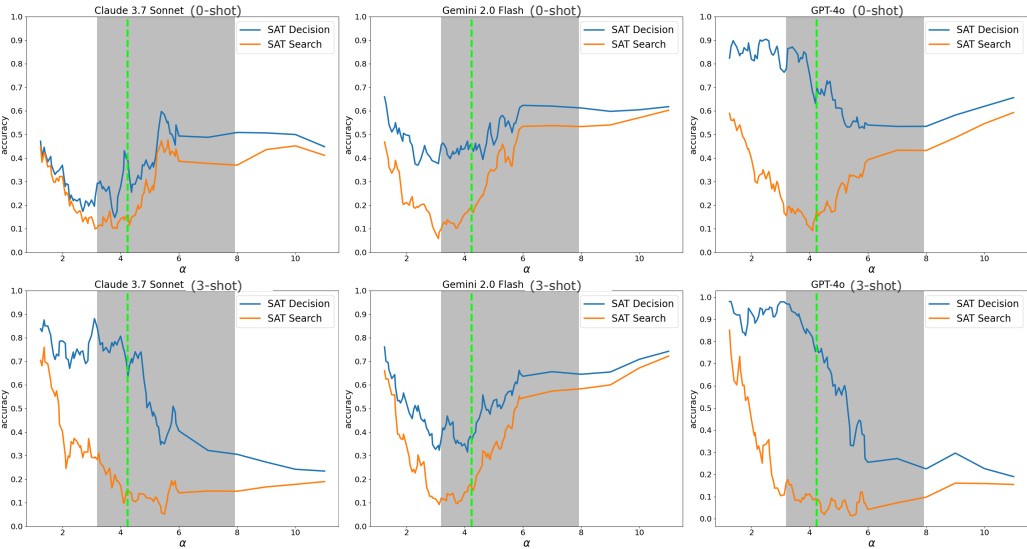

Figure 10: **0-shot vs. 3-shot accuracy.** The figure compares 0-shot and 3-shot performance comparing all LLMs. It can be observed that in-context learning does not enhance performance in the hard region. The setup SAT-Menu. The plot was generated using a size 4 moving window on $\alpha$ values.

---

**Box 5: SAT-Translate Prompt**

# System Message
You are provided with a list of preferences from different individuals, each specifying items they like and dislike. Create a logical expression in Conjunctive Normal Form (CNF) that satisfies a set of individual preferences regarding likes and dislikes of certain items. The condition for an individual's satisfaction is that either at least one item they like is included, or at least one item they dislike is excluded in your selection. Format the final CNF expression in LaTeX. Ensure all item names are retained in the final output. Do not include any explanation.

# Input for a new problem
**Preferences**: Om: Likes naan, curry. Dislikes tandoori. Bao: Likes curry. Dislikes naan, tandoori. Nic: Likes naan. Dislikes curry, tandoori. Pat: Likes curry. Dislikes naan, tandoori. Du: Likes tandoori, naan, curry. Kim: Likes curry. Dislikes tandoori, naan.

# Generated Solution for the given problem
**Solution**:

(naan $\lor$ curry $\lor$ ¬tandoori) $\land$ (curry $\lor$ ¬naan $\lor$ ¬tandoori) $\land$ (naan $\lor$ ¬curry $\lor$ ¬tandoori)$\land$

(curry $\lor$ ¬naan $\lor$ ¬tandoori) $\land$ (tandoori $\lor$ naan $\lor$ curry) $\land$ (curry $\lor$ ¬tandoori $\lor$ ¬naan)

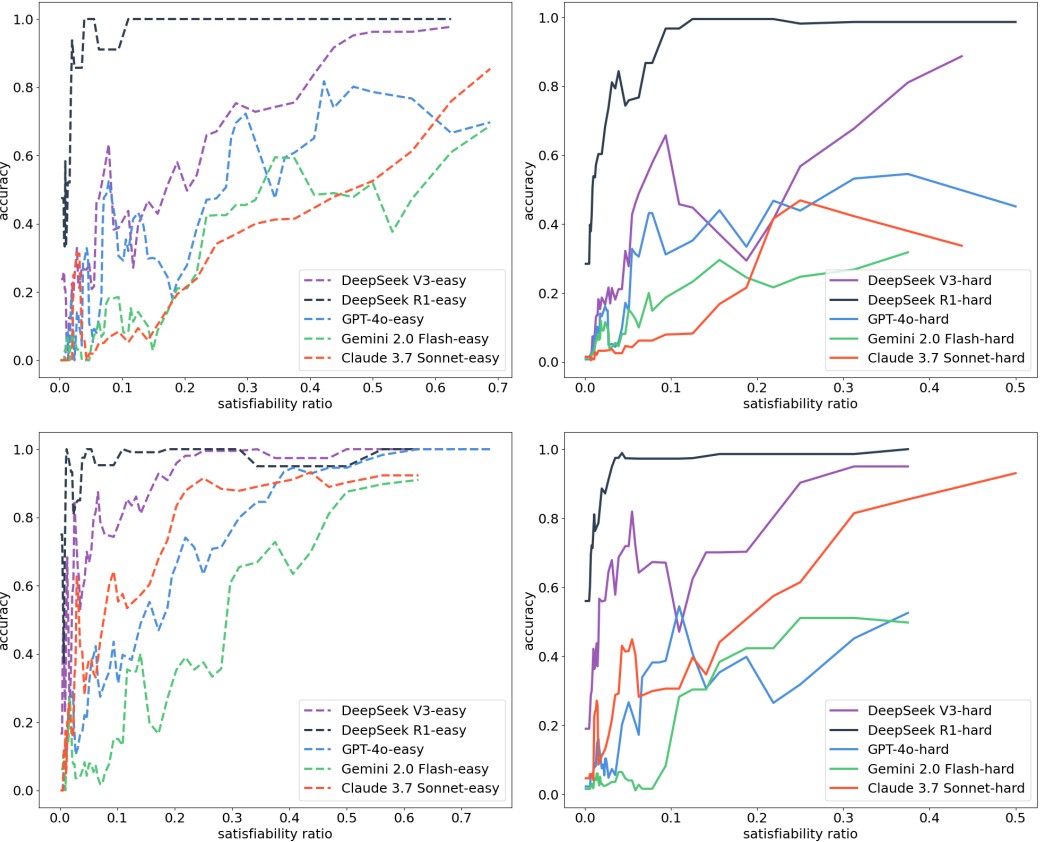

Figure 11: **Accuracy vs. satisfiability ratio.** R1 maintains consistent accuracy regardless of satisfiability ratio. In contrast, the performance of other LLMs is impacted by the number of satisfying assignments – more the satisfying assignments, higher the performance. The first row is for SAT-Menu and the second row is for SAT-CNF. We only include satisfiable instances and analyze easy (dashed line) and hard regions (solid line) separately.

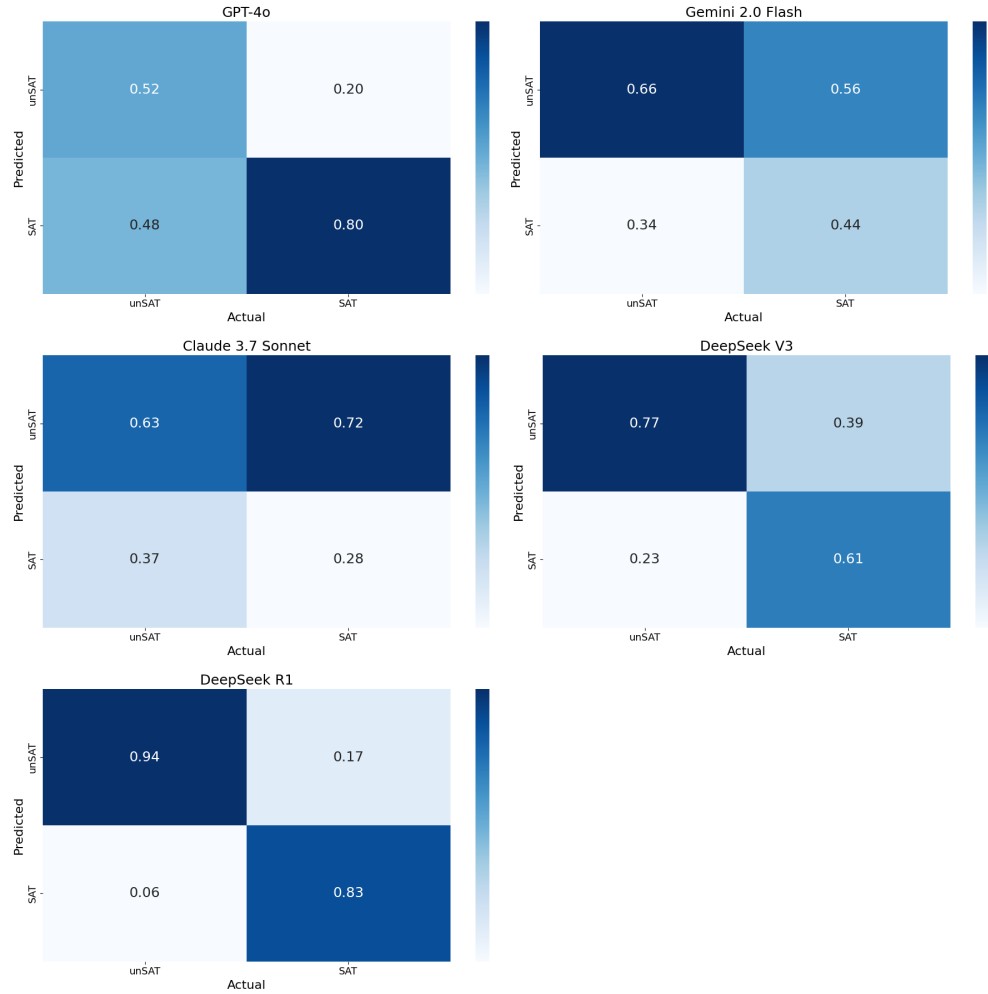

Figure 12: **Confusion matrices for the decision version of 3-SAT.** It can be observed that, except R1, all other LLMs struggle to correctly classify unsatisfiable instances. This suggests that R1 is more sound (detects sat/unsat correctly) than complete (cannot guarantee to find a solution). The cell annotations reflect classification accuracy, normalized over the true counts (column) to account for the imbalance between SAT and unSAT instances. The setup is SAT-Menu with 0-shot prompting.

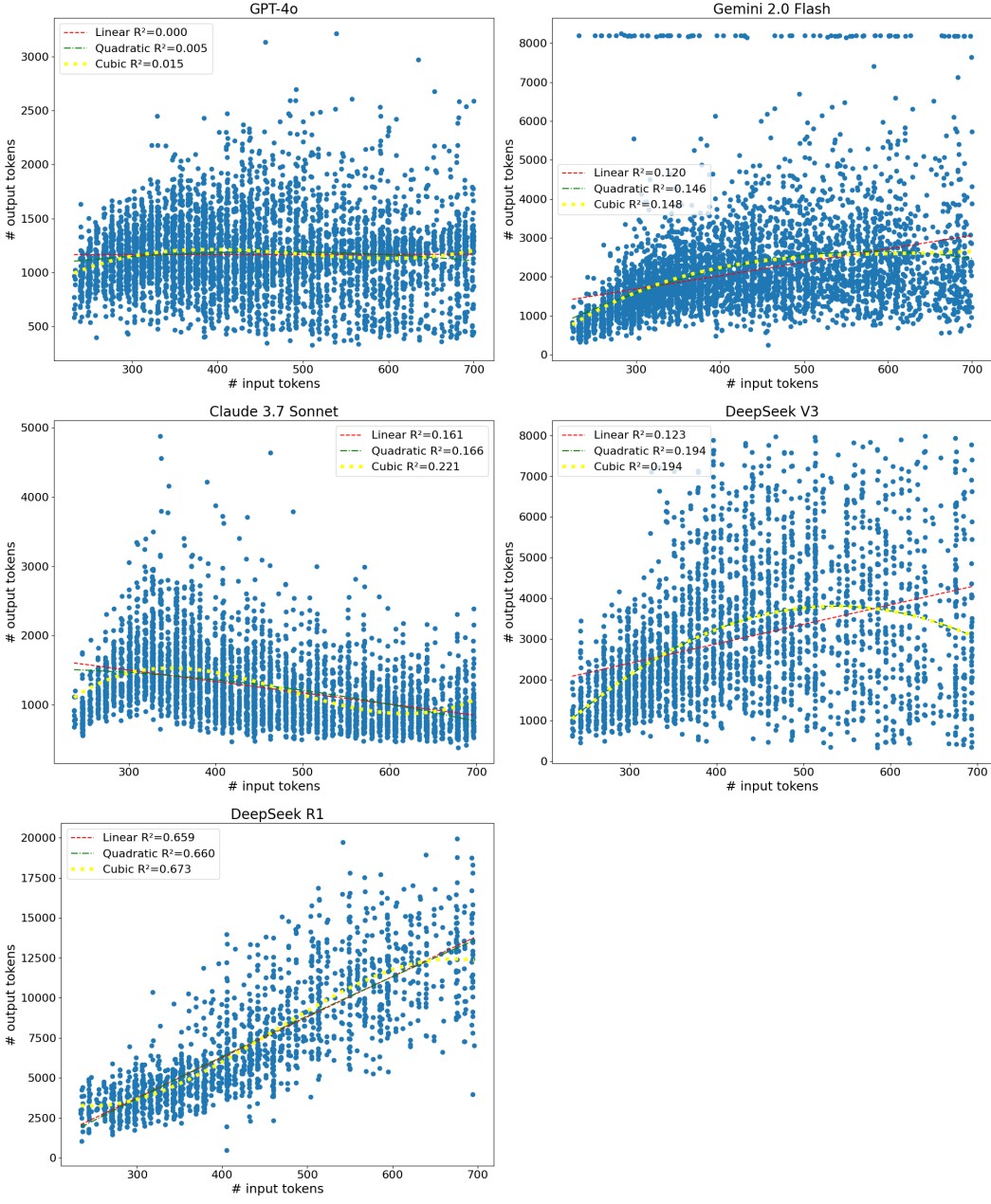

Figure 13: **Correlation between output (generated) and input tokens.** R1 output tokens grows polynomially with the input tokens. In contrast, the generated tokens of other LLMs remain largely unchanged.

