# OpenReview forum: "Have Large Language Models Learned to Reason? A Characterization via 3-SAT"
_colmweb.org/COLM/2025/Conference — COLM 2025_

### Official Review · Reviewer_jBva · 2025-05-08

**Rating:** 6
**Confidence:** 3
**Ethics Flag:** 1

**Summary:**

This paper studies the reasoning capabilities of LLMs through the lens of 3-SAT problems, leveraging the well-established phase transition behavior of 3-SAT as a proxy for problem hardness. The authors test multiple SOTA LLMs on both structured CNF representations and natural language (SAT-MENU) encodings of 3-SAT instances in decision and search formats. A key finding is that DeepSeek R1 shows behavior resembling symbolic reasoning, outperforming others in hard regions.

**Reasons To Accept:**

* The paper makes a rigorous attempt to operationalize "reasoning" by aligning it with search over NP-complete problems, providing a theoretical and empirical framework that stands out from benchmark-driven approaches.
* The study is timely and of significant interest to the NLP community, especially those working on LLM reasoning.
* Qualitative trace analyses provide insights into whether models internally simulate symbolic search strategies.

**Reasons To Reject:**

* While the setup is grounded in complexity theory, there’s no formal analysis connecting transformer architecture limitations to the observed empirical behavior. The use of complexity class boundaries is insightful but not fully exploited.
* While R1 outperforms other models, there is insufficient ablation to isolate whether the improvement comes from architecture, training data, RL fine-tuning, or test-time compute. The attribution is largely speculative.

---

> ### Author Response · Authors · 2025-06-01
> **Rebuttal to Reviewer jBva**
>
> We thank the reviewer for taking the time to review our manuscript. We hope to address any concerns through our answers.
>
> * ```No formal analysis connecting transformer architecture limitations to the observed empirical behavior```
> We respectfully point the reviewer to the theoretical guarantees stated in our Introduction (L35-48) and Related Works section which we base our work on.
>
> Specifically, we refer to (Li et al., 2024), which said that *T*-chain of thought (CoT) steps can extend transformers’ abilities beyond TC0 – With *T* being polynomial in the input size, LLMs can, **in theory, solve problems in P class like 2-SAT and Horn SAT**. However, to the best of our knowledge, **there are no empirical works that show to what extent these work in practice**, since **it is not a binary answer** but rather requires a more comprehensive study to evaluate its limits. Our SAT experiments support this, showing improved reasoning with extended CoT steps. Figures 3 and 9 (in Appendix) show that **with more CoT steps, R1 completely outperforms other LLMs that use fewer CoT steps**. Importantly, in Figure 9, **R1 performs with near-perfect accuracy on Horn-SAT and 2-SAT**, which is completely aligned with theory. In contrast, LLMs are unable to show perfect solver-like accuracy even for tractable fragments like Horn-SAT (P-Complete) and 2-SAT (NL-Complete).
>
> ---
>
> * ```Insufficient ablation to isolate whether the improvement comes from architecture, training data, RL fine-tuning, or test-time compute```
> We agree that disentangling the exact sources of performance improvement is a valuable research direction. However, the goal of our manuscript is not to pinpoint the origins of R1’s performance gains, but rather to evaluate and characterize the reasoning in LLMs and LRMs in a more principled way. **Our focus is thus on capability evaluation, not causal attribution.**
> We argue that rigorous ablation studies—while important—require access to pretraining configurations, architectures, and fine-tuning data pipelines, which are often **proprietary**. Even in *open-source settings*, **isolating individual contributions demands extensive retraining and computational resources**, which are beyond the scope of this paper and have been independently analyzed in recent works (cf., Stechly et al., 2025, Gandhi et al., 2025).
>
> **Instead, our work aims to establish a framework for reasoning evaluation and to highlight both the limitations and capabilities of current models**. We hope this foundation will support and motivate future work that explores architectural or training-related contributions through more detailed ablation.
>
>
> [1] Li et al., 2024, Chain of thought empowers transformers to solve inherently serial problems
> [2] Stechly et al., 2025, Beyond Semantics: The Unreasonable Effectiveness of Reasonless Intermediate Tokens
> [3] Gandhi et al., 2025, Cognitive Behaviors that Enable Self-Improving Reasoners, or, Four Habits of Highly Effective STaRs.

---

> > ### Comment · Reviewer_jBva · 2025-06-05
> >
> > Thank you for your response. Considering the overall quality, I will keep my original score, which was already positive. Wishing the authors the best of luck.

---

### Official Review · Reviewer_r8DN · 2025-05-09

**Rating:** 6
**Confidence:** 3
**Ethics Flag:** 1

**Summary:**

The authors present a detailed study of modern LLMs performance on a 3-SAT benchmark. By varying the complexity
level of problem instances, the authors demonstrate the dependence of accuracy of Deepseek R1, GPT-4o, Claude Sonnet and Gemini 2.0 Flash models on the hardness of the instance. Also, the authors attempt to analyze reasoning patterns of tested models. The manuscript describes similar experiments for 2-SAT and HornSAT problems.

**Questions To Authors:**

1. How many answers did you generate per problem instance? Can you provide estimates of the standard deviation for the curves in Figure 3, for example?
2. Did you perform any analysis of reasoning traces beyond empirical evaluation? For example, can you compare the frequency of SAT heuristics use by DeepSeek R1 and GPT-4o?

**Reasons To Accept:**

1. The paper is well written, the authors provide a good conceptual introduction to the problem.
2. Experimental data is convincing
3. The authors try to explain reasoning process of different LLMs.

**Reasons To Reject:**

1. Although the implementation of this study is quite good, the idea is not novel. Several works presented benchmarks for LLMs based on problems from different computational complexity classes. Specifically, https://openreview.net/pdf?id=mHx8JFURtn proposed a benchmark based on recursive problems from P class and https://arxiv.org/abs/2312.14890 proposed multiple benchmarks based on P, NP-complete and NP-hard problems, to name a few. The authors should improve the overview of the existing works.
2. The authors do not go beyond simple model prompting, although there are multiple test-time techniques to improve LLM reasoning (Tree-of-Thought, Program-of-Thought etc). Comparing the performance of reasoning models to at least one of these techniques is important to understand the weaknesses of current models.

---

> ### Author Response · Authors · 2025-06-01
> **Rebuttal to Reviewer r8DN**
>
> We thank the reviewer for taking the time to review our paper and giving us detailed feedback. We hope to answer your questions and clarify any doubts here.
>
> * ```Although the implementation of this study is quite good, the idea is not novel.```
> Thank you for pointing out these works. We note that NPHardEval (Fan et al., 2024) was included in an earlier draft of our manuscript, but was unintentionally omitted during revisions. We will reintegrate it into the revised version.
> *Closest to our work is the NPHardEval (Fan et al. (2024)), which examines reasoning across various computational complexity classes, but we additionally explore phase transition characteristics and investigate how the performance varies with inherent hardness of problems. Our evaluation centers on a range of complexity classes known to exhibit phase transitions (Schaefer 1978) including 2-SAT (NL-Complete), 1-3 Horn-SAT (P-Complete), and 3-SAT (NP-Complete). Our comparison of LRMs (R1) with LLMs provide better insights into the effectiveness of longer reasoning traces and establishes better reasoning abilities on the easier and hard regions – and how performance varies with model count (i.e. number of solutions) of a problem.*
>
> The same distinctions apply to “RETHINKING LOGIC IN AI” (Nechesov et al., 2025) which is mainly restricted to P-class. That said, **the latter manuscript appears to remain unpublished in a peer-reviewed venue** (including a recent rejection from ICLR 2025) and is not currently available on preprint servers such as arXiv.
>
> ---
>
> * ```Test-time techniques like Tree-of-thoughts nor evaluated```
> As stated in Lines 170–177 (including Footnote 3), our primary objective is to evaluate the **native reasoning capabilities of *autoregressive* LLMs, independent of external scaffolding or symbolic wrappers**.
>
> 1. There are **theoretical bounds for autogressive LLMs** in what they can compute (cf., L 35-48).  Investigating their reasoning capabilities in isolation allows us to better understand these inherent bounds without confounding effects introduced by external scaffolds; This is exactly why we evaluate models which we know are autoregressively generating tokens like R1  – and not models like o1 for which we cannot determine the same.
> 2. Techniques such as Tree-of-Thoughts, Best-of-N sampling, or other tree-based search frameworks effectively wrap LLMs in an external symbolic search. These methods use the LLM to generate ideas (nodes of the tree) while orchestrating search or backtracking over a tree data structure. **Crucially, the graph algorithm (e.g., tree traversal, backtracking) is not *internally* generated or maintained by the LLM. Instead, it is *externally* imposed**. Our goal is to assess whether LLMs can natively reason — for example, **whether they can internally generate multiple candidate solutions, maintain them in memory, reason over them, and converge to a consistent answer**, *within the constraints of autoregressive generation* (Figures 4, 7). Even Best-of-N sampling — often treated as a minimal enhancement — effectively introduces a single-layer tree where the LLM samples multiple independent completions. This is still not native reasoning, as the LLM does not internally track or compare alternatives.
> 3. While we acknowledge that test-time techniques can significantly improve performance, recent literature suggests their **effectiveness is tied to the reasoning ability of the base model** (Gandhi et al., 2025). Therefore, it is crucial to test these baseline abilities.
>
> We will add these points to the manuscript for further clarification for the readers.
>
>
> [1]  Fan et al., 2024, NPHardEval: Dynamic Benchmark on Reasoning Ability of Large Language Models via Complexity Classes
> [2]  Nechesov et al., 2025, Rethinking logic in AI: A novel benchmark inspired by polynomial analogue of Gandy's fixed point theorem
> [3] Schaefer, T. J. (1978,). The complexity of satisfiability problems. In Proceedings of the tenth annual ACM symposium on Theory of computing.
> [4] Gandhi et al., 2025. Cognitive Behaviors that Enable Self-Improving Reasoners, or, Four Habits of Highly Effective STaRs.

---

> > ### Author Response · Authors · 2025-06-01
> > **Rebuttal to Reviewer r8DN (Questions)**
> >
> > * ```How many answers did you generate per problem instance? Can you provide estimates of the standard deviation for the curves in Figure 3, for example?```
> >
> > We generate only 1 answer with a very low temperature (~0.3) for every problem instance. This is again linked to our explanation of Best-of-N above and why we only focus only on the native reasoning abilities of autoregressive LLMs.
> >
> > We have now added **confidence intervals** to Figures 3 and 9. We use 90% confidence intervals to quantify uncertainty around the estimated accuracy. Each interval shows the range in which the true accuracy is likely to lie with 90% confidence. In the plots, these intervals help assess the reliability of accuracy estimates—narrow bands indicate high certainty, while wider bands suggest greater variability. The figures are available via this anonymous link: https://anonymous.4open.science/r/COLM-rebuttal-figures-95E3/README.md
> >
> > We will revise our paper to update these figures. We thank the reviewer for their suggestion.
> >
> > ---
> >
> > * ```Did you perform any analysis of reasoning traces beyond empirical evaluation? For example, can you compare the frequency of SAT heuristics used by DeepSeek R1 and GPT-4o?```
> > Conducting such an analysis presents significant practical and theoretical challenges.
> > (1) **Reliably parsing LLM-generated natural language outputs** to extract exact frequencies of heuristic usage **is theoretically non-trivial**. Most available tools for automated parsing – such as regular expressions – operate within the confines of Type 3 or Type 4 grammars in the Chomsky hierarchy. In contrast, natural language generation by LLMs corresponds to a Type 0 language, requiring the full expressive power of a Turing machine to accurately parse and interpret. This discrepancy makes rule-based extraction brittle and incomplete.
> > (2) **Identifying heuristic usage assumes a fixed, human-curated inventory of heuristics**. This approach may be fundamentally limited, as LLMs often exhibit emergent behaviors that deviate from canonical SAT-solving strategies. They may invent or blend heuristics in ways that are difficult to map to standard categories, especially without strong interpretability tools. A concrete example of this is shown in Figure 7 (Appendix), where DeepSeek R1 appears to combine local search and backtracking—two strategies originating from distinct search paradigms—within a single reasoning trace.
> > (3) Finally, the **sheer complexity of a single R1 trace is 25-30 pages long**, so unless the models’ generation is restricted to predefined structures through training (cf., Stechly et al., 2025), we cannot apply formal methods to perform a fine-grained analysis of these methods.
> >
> >
> > [5] Stechly et al., 2025, Beyond Semantics: The Unreasonable Effectiveness of Reasonless Intermediate Tokens

---

### Official Review · Reviewer_ApVz · 2025-05-12

**Rating:** 6
**Confidence:** 1
**Ethics Flag:** 1

**Summary:**

This paper investigates the reasoning ability of large language models (LLMs) by evaluating their performance on 3-SAT problems, a well-established problem in complexity theory. The paper compares the models’ performance under two problem presentation modes (SAT-CNF and SAT-Menu) and two task types (SAT Decision and SAT Search). Results are compared with the known satisfiability threshold of random 3-SAT. The paper contributes to understanding how LLMs perform on complex reasoning tasks and explores their capabilities in problem-solving.

**Reasons To Accept:**

1. The paper explores an interesting and original approach by applying LLMs to complex combinatorial problems (3-SAT), which is significant for understanding the reasoning capabilities of these models.
2. The paper clearly defines the task, using well-established benchmarks like 3-SAT to evaluate LLMs, providing clarity in the experimental setup and comparisons.
3. The analysis of how different modes of presenting SAT problems to LLMs affects their performance is valuable, especially considering the growing interest in using LLMs for complex problem-solving.

**Reasons To Reject:**

Drawing the conclusion that “Large Language Models have Learned to Reason” solely based on 3-SAT problems seems a bit far-fetched. I hope the authors can use a more appropriate phrasing.

---

> ### Author Response · Authors · 2025-06-01
> **Rebuttal to Reviewer ApVz**
>
> We thank the reviewer for their positive evaluation of our paper. We hope to clarify the scope of our claims here
>
> ```Drawing the conclusion that *Large Language Models have Learned to Reason* solely based on 3-SAT problems seems a bit far-fetched```
> We respectfully clarify that our claim is more nuanced: we **DO NOT assert that LLMs have learned to reason**. Rather, we argue that Large Reasoning Models (LRMs) such as **R1 exhibit a step-change in reasoning ability compared to previous LLMs** (cf., Conclusion section). Our conclusion is not based solely on performance over **3-SAT** instances, but rather grounded in a broader, theoretically informed framework for evaluating reasoning **across different complexity classes** and problem types, including **2-SAT** and **Horn-SAT**.
>
> 1. 3-SAT is not an arbitrary task—it is the prototypical NP-Complete problem in complexity theory. As discussed in AI literature, including by Russell & Norvig [2] and Bottou [1], reasoning is characterized by the capacity to solve problems that require compositional, multi-step inference and search. 3-SAT satisfies all these properties, and other (propositional fragments of) reasoning problems such as **logic inference, AI planning, and constraint satisfaction, can be mapped to 3-SAT**. Thus, it serves as a principled and computationally meaningful testbed for evaluating reasoning capabilities.
> 2. Our evaluation is not limited to NP-complete problems. We also extend our analysis to problems in lower complexity classes—such as 2-SAT (NL-complete) and 1-3 Horn-SAT (P-complete)—as shown in Figure 9. The fact that **R1 achieves near-perfect accuracy on tractable classes and maintains stability in the hard region of 3-SAT** strongly suggests the learning of structured reasoning behavior rather than just fitting on statistical patterns like GPT-4o, Gemini 2.0, and Claude 3.7 Sonnet.
>
> In summary, our claim is not that LLMs have universally solved reasoning, but rather that models like R1 exhibit a step-change in reasoning behavior when evaluated under well-defined computational criteria – that is suggestive of them **having learned the reasoning rather than fitting on statistical patterns**. Our results are also **aligned with theoretical limits of transformer architecture** (Li et al., 2024), where T-CoT steps (T being polynomial in input)  can make transformers strictly more expressive. The methodology and results are aligned with theoretical definitions of reasoning.
>
> That said, as noted in our Conclusion, our 3-SAT benchmarks are bounded (10 variables, 110 clauses), and R1 still struggles in the Hard region. Thus, we acknowledge that R1 is still far from mastering reasoning in a general sense. We will revise the conclusion to better reflect these limitations and further clarify the scope of our claims.
>
> [1] Bottou, Léon, 2014. "From machine learning to machine reasoning: An essay."
> [2] Stuart Russell and Peter Norvig, 2010. "Artificial Intelligence: A Modern Approach."
> [3] Li et al., 2024, Chain of thought empowers transformers to solve inherently serial problems

---

> > ### Comment · Reviewer_ApVz · 2025-06-05
> >
> > Thank you for your response. All my questions have been resolved. I will maintain my current score, which is marginally above the acceptance threshold.

---

### Official Review · Reviewer_YUcH · 2025-05-12

**Rating:** 6
**Confidence:** 4
**Ethics Flag:** 1

**Summary:**

This paper test if LLMs reason genuinely like human or answer questions with shortcuts. To do that, the authors select the 3-SAT questions for the evaluation. Results show that performance of LLMs drops significantly when the question becomes difficult, indicating the shortcut manners. While R1 shows the manner of performing real reasoning.

**Reasons To Accept:**

1. The research question is important. Researching the reasoning manners is helpful to understand the limitations and strengths of reasoning models (e.g., R1), which could be inspiring for the further improvement.

2. The evaluation task is suitable. 3-SAT questions are NP-complete, which might be helpful to alleviate the shortcut issues when testing reasoning ablitiy.

**Reasons To Reject:**

My main concerns are about the experiment setting and the performance interpretation.

1. In previous work that research the limitation of transformer architectures, the researched models are not pretrained models, since the pretraining corpus and training processes are not that clean. The data leaky may happen and this would influence the final evaluation performance. However, in this work, the authors research the limitations mainly for pretrained models, which inevitably encounters this issue. It's not very convincing to connect the evaluation results with the real reasoning ability under this condition.

2. Based on the reported performance (e.g., Figure 3), it's not very obvious to conclude that R1 are more like true reasoning. The tendency is similar to those of other LLMs, and the performance gap between it and other models is roughly stable. Besides, some case study without statistical support is still not sufficient to prove the advantage of reasoning models w.r.t. the LLMs.

---

> ### Author Response · Authors · 2025-06-01
> **Rebuttal to Reviewer YUcH**
>
> We'd like to thank the reviewer for taking the time to review our paper and asking interesting questions.
>
> * ```Evaluation of reasoning abilities of pretrained models that may have data leakage issues```
> If we understand correctly, the reviewer suspects that SAT data might be present in the training corpus of the pretrained models. If so, this is a great question that we intend to clarify, but we might have fallen short in the paper.
>
> We, too, suspect that ALL models have likely encountered SAT data during pretraining. This is evidenced by their immediate use of terminology and strategies (e.g., tree search, backtracking) when presented with SAT instances in CNF form. **However, the core concern is not whether SAT-related content exists in the training data** (which it likely does), **but whether our specific evaluation is compromised by data leakage.**. We address this by considering two scenarios:
>
> 1. **Exact data overlap**.  The strongest form of data leakage occurs when the evaluation set contains identical instances present in the model’s training data. This scenario is highly unlikely in our case. **We use a custom SAT data generator, and neither the data nor the generator has been open-sourced or publicly released**. This rules out direct memorization.
>
> 2. **In-distribution, but non-identical data.** The most plausible scenario is that the evaluation data shares a distributional similarity with pretraining data. Even in this case, our use of 3-SAT—particularly instances from the hard region—minimizes reliance on shallow statistical patterns. In the easy region, LLMs can exploit superficial heuristics (e.g., associating longer input lengths with unsatisfiability, or randomly assigning variables in the satisfiable Easy region). However, **in the hard region, such patterns break down, and models must engage in actual multi-step reasoning and search to succeed**. This is precisely why we focus on these harder instances.
>
> A third possibility is **Out-of-distribution generalization**, where models must rely on reasoning rather than statistical shortcuts. This is consistent with prior work (Zhang et al., 2022; Dziri et al., 2023), which demonstrates that LLMs trained on reasoning tasks suffer sharp performance degradation under even modest distribution shifts when fitting on statistical patterns.
>
> **We also highlight our SAT-Menu setup**, where the problem is reframed as a menu-based preference selection task. It is extremely unlikely that such a format appears in pretraining data, yet we still observe a clear performance gap between R1 and other models (see Figures 3,7).
>
> ---
> * ```Based on the reported performance (e.g., Figure 3), it's not very obvious to conclude that R1 is more like true reasoning.```
> > ```Tendency similar to other LLMs and the performance gap is roughly stable```
> We emphasize that our claim is **NOT that R1 has achieved complete reasoning ability** (see Conclusion), but that it exhibits a **step-change compared to other LLMs**. This is not merely based on its performance on **3-SAT** (Figure 3), but also its near-perfect performance on **2-SAT** (NL-Complete) and **1-3 Horn-SAT** (P-Complete) as shown in Figure 9 (in Appendix). Importantly, the stability of R1’s performance across both the easy and hard regions in Figure 3 – and the fact that its hard region performance surpasses even the easy region performance of other LLMs – **is not a limitation—it is precisely what characterizes a step-change in reasoning.***
> Moreover, we show that the performance remains stable regardless of the number of solutions of a problem instance (Fig. 3 right, 11).
> Our empirical results are further justified by the qualitative analysis of the reasoning traces of R1 and GPT-4o that show that R1 indeed performs a more consistent tree search in an autoregressive manner.
> **Taken together—strong empirical results, consistent behavior traces, and theoretical limits of transformer architecture** (Li et al., 2024 and Peng et al., 2024) — **we argue that R1 marks a meaningful advancement in the reasoning abilities of LLMs.**
>
> > ```Statistical significance missing```
> We have now added **confidence intervals** to Figures 3 and 9. We use 90% confidence intervals to quantify uncertainty around the estimated accuracy. In the plots, these intervals help assess the reliability of accuracy estimates—narrow bands indicate high certainty, while wider bands suggest greater variability. The figures are available via this anonymous link: https://anonymous.4open.science/r/COLM-rebuttal-figures-95E3/README.md
>
> We thank the reviewer for their suggestion and will include it in the revised version.
>
> [1] Zhang et al., 2022, On the paradox of learning to reason from data
> [2] Dziri et al., 2023, Faith and fate: Limits of transformers on compositionality
> [3] Li et al., 2024, Chain of thought empowers transformers to solve inherently serial problems
> [4] Peng et al., 2024, On limitations of the transformer architecture

---

> > ### Comment · Reviewer_YUcH · 2025-06-01
> > **Response to the authors' reply**
> >
> > Thanks for the detailed explanation. The response addressed my concern. Perhaps the authors could include this in the main paper. I'll update my score accordingly.

---

### Decision · Program_Chairs · 2025-07-08

**Decision:**

Accept

**Comment:**

The paper investigates whether large language models are able to truly reason or instead rely on shortcuts by evaluating their performance on 3-SAT problems. Experimental results indicate that performance drops as problem difficulty increases. Based on the results, the authors notice that DeepSeek R1 may showcase behavior that is close to symbolic reasoning, while other models do much worse for harder instances. The reviewers appreciated the research question and experimental design. The reviewers also raised concerns about data leakage from pretrained models and the clarity of claims regarding genuine reasoning. The authors clarified that direct data overlap is unlikely due to custom data generation, and that focusing on hard 3-SAT instances reduces reliance on superficial patterns. The authors also noted that R1’s performance is stable across complexity classes and supplemented empirical results with qualitative analysis of reasoning traces. Reviewers suggested more nuanced language and a stronger overview of related work, to which the authors responded by emphasizing their focus on evaluating native reasoning rather than attributing improvements to specific factors. The authors added confidence intervals to figures to address statistical concerns and agreed to clarify claims and limitations in the revised manuscript. Overall, the reviewers and the area chair found the work valuable and well-executed but recommended additional work, which the authors committed to addressing in the final version of the manuscript. The reviewers seemed satisfied with the authors’ responses and maintained their positive evaluations. The area chair would like to second the reviewers’ comments and suggestions to nuance the conclusions. In particular, the term “characterization:” in the title and in the text may be a bit too strong; “investigation” may be more appropriate here.